# ArcDAE: Asymmetric Rectified Contrastive Diffusion Autoencoder for Unified Representation Learning

**Ge Gao** [1]  **Di Xiong** [1]  **Zeke Xie** [2]  **Jian Yang** [1 3]  **Shuo Chen** [* 1]

## Abstract

The unification of generative details and discriminative semantics presents a structural paradox in *diffusion-based representation learning*. Early approaches decouple semantics from generation, inevitably compromising representational completeness (i.e., *information split*). While recent bridge-based methods achieve unification via a tightly coupled mapping, they suffer from *information overload*. This is because unconstrained reconstruction objectives incentivize the encoder to entangle high-frequency stochastic noise into the latent bottleneck. To solve this, we introduce *asymmetric rectified contrastive diffusion autoencoder* (ArcDAE), which rebuilds the diffusion bridge as a *dynamic sifter*. Through imposing a *timestep-aware rectification constraint* that orthogonalizes the semantic manifold from the stochastic noise space, ArcDAE compels the bottleneck to distill discriminative features while actively shedding high-frequency redundancy. Consequently, our approach eliminates the overload trap without reverting to decoupling. Extensive experiments validate the superiority of our FFHQ-trained ArcDAE, surpassing state-of-the-art methods by up to 6.4% in downstream semantics regression and 9.7% in reconstruction fidelity.

## 1. Introduction

Unifying generative fidelity and discriminative utility presents a paradox in unsupervised learning (Kingma & Welling, 2014; Abdal et al., 2019; 2020; Preechakul et al., 2022). Since discrimination seeks invariance while diffu-

sion demands variance for reconstruction (Ho et al., 2020; Dhariwal & Nichol, 2021; Zhang et al., 2022), balancing sufficient generative detail with noise suppression remains challenging (Wang et al., 2023; Yang et al., 2023; Yue et al., 2024; Wu & Zheng, 2024).

To bridge this gap, the field has evolved through a dialectical process oscillating between two structural extremes, as illustrated in Figure 1. Early approaches (Preechakul et al., 2022; Zhang et al., 2022; Wang et al., 2023) adopted an *information split* strategy (Figure 1(a)). By utilizing a separate auxiliary encoder for semantics while relegating high-frequency details to an independent stochastic chain, they achieved disentanglement but at the cost of decoupling the semantic latent from the generative driver which limits representational completeness. Conversely, recent diffusion bridge paradigms (Zhou et al., 2023; He et al., 2024; Ji et al., 2024; Kim et al., 2025; Sanokowski et al., 2025) remove the auxiliary path to enforce a bijective mapping between the latent bottleneck and the noisy diffusion endpoint (Figure 1(b)).

While these bridge-based methods ensure information completeness, we identify that they inadvertently trigger an *information overload* which manifests specifically as *asymptotic noise dominance*. In an unconstrained diffusion bridge, minimizing the reconstruction error forces the latent bottleneck to act as a sufficient statistic for the terminal state (Ji et al., 2024; Kim et al., 2025; Sanokowski et al., 2025). Since the terminal state converges to an isotropic Gaussian distribution representing pure noise as the diffusion process proceeds (Sohl-Dickstein et al., 2015; Song et al., 2021b), the encoder is compelled to prioritize high-entropy stochastic noise over lower-entropy semantic structure. Consequently, the latent space degenerates into a non-discriminative container of noise entropy, thereby negating the benefits of the unified formulation and inflexible dimensionality (Oksendal, 2013; Sinha et al., 2021).

To prevent the latent space from degenerating into a noise map, we argue that a robust encoder must function as an *information sifter* rather than a passive reconstruction bridge. We introduce the *asymmetric rectified contrastive diffusion autoencoder* (ArcDAE) which employs an asymmetric protocol where the input is structurally degraded yet the target

[1] School of Intelligence Science and Technology, Nanjing University [2] Information Hub, The Hong Kong University of Science and Technology (Guangzhou) [3] School of Computer Science and Technology, Nanjing University of Science and Technology. Correspondence to: Shuo Chen <shuo.chen@nju.edu.cn>.

*Proceedings of the $43^{rd}$ International Conference on Machine Learning*, Seoul, South Korea. PMLR 306, 2026. Copyright 2026 by the author(s).

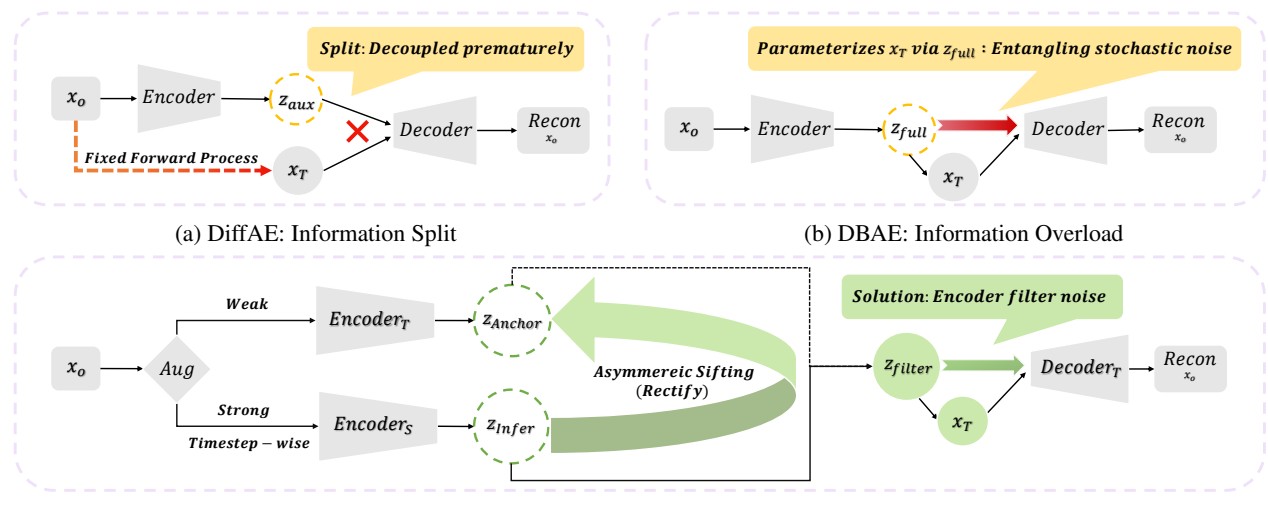

*Figure 1.* A quick comparison among the conventional approaches, the bridge-based approaches, and our proposed method. (a) The *information split* in *diffusion autoencoder* (DiffAE) (Preechakul et al., 2022). Due to the fixed forward, the input $\mathbf{x}_0$ is bifurcated into an auxiliary latent ($\mathbf{z}_T$) and an independent diffusion endpoint ($\mathbf{x}_T$), leading to decoupled representations. (b) The *information overload* in *diffusion bridge autoencoder* (DBAE) (Kim et al., 2025). Driven by *unconstrained reconstruction*, $\mathbf{z}_{\text{full}}$ becomes a *sufficient statistic* for the noisy $\mathbf{x}_T$, inherently entangling high-frequency stochastic noise and diluting discriminability. (c) Our ArcDAE builds an asymmetric teacher-student architecture to actively rectify the noisy inferred latent ($\mathbf{z}_{\text{infer}}$) towards a clean semantic anchor ($\mathbf{z}_{\text{anchor}}$).

remains semantically stable (Figure 1(c)). This forces the model to abandon noise memorization and instead learn to rectify corrupted inputs based on learned global semantic priors (Grill et al., 2020; Caron et al., 2021).

Our contributions are three-fold:

- We introduce ArcDAE, a unified framework that solves the dichotomy between *information split* and *information overload* by reconfiguring the diffusion bridge as a dynamic sifter to filter high-frequency noise while preserving semantic density.
- We propose *asymmetric semantic rectification* (ASR) to actively orthogonalize the semantic latent from the noise manifold, complemented by *timestep-adaptive granularity alignment* (TGA) to ensure structural integrity across the diffusion trajectory.
- We demonstrate that ArcDAE surpasses state-of-the-art methods by up to 6.4% in downstream semantic tasks and 9.7% in reconstruction fidelity, establishing a new Pareto frontier for unified representation learning.

## 2. Related Work

We review diffusion representation learning via information coupling and discuss how contrastive optimization grounds our rectification paradigm.

### 2.1. Diffusion Representation Paradox

Diffusion probabilistic models have redefined generative fidelity (Sohl-Dickstein et al., 2015; Ho et al., 2020; Rombach

et al., 2022). Typically formulated via stochastic differential equations (Anderson, 1982; Oksendal, 2013; Song et al., 2021b) or score matching (Vincent, 2011; Song & Ermon, 2019), these models degrade data into noise and learn to reverse the process. Despite the availability of deterministic sampling trajectories (Song et al., 2021a), utilizing this generative backbone for discrimination presents a structural challenge known as the *fidelity-utility paradox*.

**From Information Split to Diffusion Bridges.** Early approaches utilizing auxiliary encoders suffer from an *information split* where semantic details dissolve into uncontrollable noise (Preechakul et al., 2022; Zhang et al., 2022; Yue et al., 2024). To ensure completeness, recent research parameterizes the diffusion endpoint directly via the latent bottleneck (Zhou et al., 2023; Ji et al., 2024; Kim et al., 2025; Sanokowski et al., 2025) and incorporates consistency constraints (Song et al., 2023; He et al., 2024). Although resolving the split, these unconstrained bridges induce an *information overload trap* by mandating the bottleneck to reconstruct high-frequency noise and compelling the encoder to prioritize noise memorization over semantic abstraction.

**Feature Alignment and Distillation.** Studies validate the emergence of semantic correspondence within internal diffusion features (Baranchuk et al., 2022; Hedlin et al., 2023; Luo et al., 2023; Hedlin et al., 2024; Tian et al., 2024). Subsequent strategies optimize feature utilization by aligning internal maps (Zhang et al., 2023; Chen et al., 2025b) or distilling generative capabilities into discriminative backbones (Li et al., 2023; Yu et al., 2024). While enhancing quality, these methods leave the structural noise entanglement of

the bottleneck unresolved. ArcDAE conversely addresses this bridge-based overload by redefining the objective as *asymmetric semantic rectification* to achieve a balanced and noise-rectified representation.

## 2.2. Towards Information Balanced with Rectification

**Invariance and Asymmetry.** Contrastive learning secures invariance through instance discrimination (Chen et al., 2020; He et al., 2020; Wang & Isola, 2020), hard negative mining (Robinson et al., 2021), and regularization (Bardes et al., 2022). Beyond pairwise comparisons, clustering mechanisms capture semantic structure (Caron et al., 2020; Li et al., 2021) while asymmetric architectures stabilize learning via momentum teachers (Grill et al., 2020; Chen & He, 2021; Caron et al., 2021). ArcDAE exploits this asymmetry by employing the teacher as a semantic anchor to rectify noisy student inputs and filter the redundancy.

**Granularity and Decoupling.** Multi-granular frameworks emphasize managing diverse information scales for dense prediction (Wang et al., 2021; Xie et al., 2021; Zhou et al., 2022). Recent studies relax strict margins (Sun et al., 2020; Wei et al., 2021) to handle ambiguity via similarity-agnostic learning (Chen et al., 2025a) or progressive augmentation (Tan et al., 2025). Integrating these principles, our *timestep-adaptive granularity alignment* (TGA) leverages orthogonality between holistic and partial representations. TGA diverges from prior focus on feature richness by ensuring noise filtering preserves structural integrity for an information-balanced state.

# 3. Method

We propose ArcDAE to address the *information overload problem* inherent in diffusion bridges. Intuitively, standard unconstrained diffusion bridges often act as *passive sponges*, absorbing an unfiltered mixture of both semantic *essence* and stochastic *residue*. Crucially, our objective is not simply to over-filter this mixture into featureless distilled water to isolate bare semantics. Instead, we aim to precisely sift out the stochastic sediment (useless noise) while retaining as much structural texture (high-frequency details) as possible.

Unlike standard diffusion ending in Gaussian noise, bridges enforce pixel-level boundary conditions, compelling the model to encode high-frequency nuisance variables alongside semantics. By reframing the bridge as an asymmetric architecture, ArcDAE functions as a *dynamic information sifter* (Figure 2), utilizing *asymmetric semantic rectification* (ASR) to filter information flow and *timestep-adaptive granularity alignment* (TGA) to govern latent granularity. This design ensures that ArcDAE preserves both low-frequency global semantics and high-frequency structural details, thereby preventing information overload and establishing a superior balance between semantic representation

and fine-grained reconstruction fidelity.

## 3.1. Problem Analysis: The Score Matching Dilemma

**Preliminaries and Notation.** Let the data distribution $p_{\text{data}}$ be supported on a low-dimensional differentiable manifold $\mathcal{M} \subset \mathbb{R}^D$ with intrinsic dimension $d_{\mathcal{M}}$, consistent with the classical embedding theory (Whitney, 1936). We use $D$ only for the ambient data dimension, $d_{\mathcal{M}}$ for the intrinsic manifold dimension, $d_z$ for the latent bottleneck dimension, and $d_h$ for the representation dimension. We formally distinguish between the continuous-time stochastic process $(\mathbf{X}_t)_{t \in [0,T]}$ and its realization $\mathbf{x}_t \in \mathbb{R}^D$. The bridge process transitions from raw data $\mathbf{X}_0 \sim p_{\text{data}}$ to a target random variable $\mathbf{X}_T = \text{Dec}(\mathbf{z})$, where $\mathbf{z} \sim \mathcal{N}(\mathbf{0}, \mathbf{I}_{d_z})$ is a latent variable and $\text{Dec} : \mathbb{R}^{d_z} \to \mathcal{S}$ is a decoder mapping to the submanifold $\mathcal{S} \subset \mathcal{M}$. We use $\mathbf{x}_T$ to denote a fixed realization of $\mathbf{X}_T$ when conditioning the bridge.

**Bridge Dynamics.** The diffusion is governed by an $h$-transformed *stochastic differential equation* (SDE). We explicitly define the components as follows: let $\mathbf{f}(\mathbf{x}, t) : \mathbb{R}^D \times [0, T] \to \mathbb{R}^D$ be the prior drift, $g(t) \in \mathbb{R}$ be the diffusion coefficient, and $\mathbf{W}_t$ denote the standard $D$-*dimensional wiener process* (Kim et al., 2025). Defining the conditional score $\mathbf{s}^*(\mathbf{x}, \mathbf{x}_T) := \nabla_{\mathbf{x}} \log p(\mathbf{x}_T \mid \mathbf{x})$, the bridge SDE is given by:

$$d\mathbf{X}_t = \left[ \mathbf{f}(\mathbf{X}_t, t) + g(t)^2\, \mathbf{s}^*(\mathbf{X}_t, \mathbf{x}_T) \right] dt + g(t)\, d\mathbf{W}_t. \quad (1)$$

**Filtration-adapted Bridge Construction.** In Eq. (1), the endpoint $\mathbf{x}_T = \text{Dec}(\mathbf{z})$ is fixed before simulating the bridge conditional on the sampled latent $\mathbf{z}$, and is therefore treated as an $\mathcal{F}_0$-measurable boundary parameter under the enlarged filtration $\widetilde{\mathcal{F}}_t = \mathcal{F}_t \vee \sigma(\mathbf{x}_T)$ rather than as an anticipating future variable. The drift can be understood as a Doob $h$-transform, so $\mathbf{s}^*(\mathbf{X}_t, \mathbf{x}_T) = \nabla_{\mathbf{X}_t} \log p_{T|t}(\mathbf{x}_T \mid \mathbf{X}_t)$ is $\widetilde{\mathcal{F}}_t$-measurable and non-anticipating with respect to the enlarged filtration. A more detailed explanation is included in Appendix A.1. It is important to note that the reference process (without the bridge term $\mathbf{s}^*$) induces a transition density denoted by $p_{0t}(\mathbf{x}_t \mid \mathbf{x}_0)$. Strictly speaking, the conditional random variable follows a Gaussian distribution:

$$(\mathbf{X}_t \mid \mathbf{X}_0 = \mathbf{x}_0) \sim \mathcal{N}(\alpha_t \mathbf{x}_0, \sigma_t^2 \mathbf{I}_D), \quad (2)$$

where $\alpha_t$ and $\sigma_t$ are the noise schedule coefficients derived from $\mathbf{f}$ and $g(t)$.

**Geometric Decomposition.** To analyze the learning objective, consider the geometry of $\mathcal{M}$. For any $\mathbf{x} \in \mathcal{M}$, the ambient space decomposes into $\mathbb{R}^D = \mathcal{T}_{\mathbf{x}} \mathcal{M} \oplus \mathcal{N}_{\mathbf{x}} \mathcal{M}$, where $\mathcal{T}_{\mathbf{x}} \mathcal{M}$ and $\mathcal{N}_{\mathbf{x}} \mathcal{M}$ denote the tangent and normal spaces, respectively. Let $\mathbf{\Pi}_{\mathbf{x}} \in \mathbb{R}^{D \times D}$ be the orthogonal projection operator onto $\mathcal{T}_{\mathbf{x}} \mathcal{M}$.

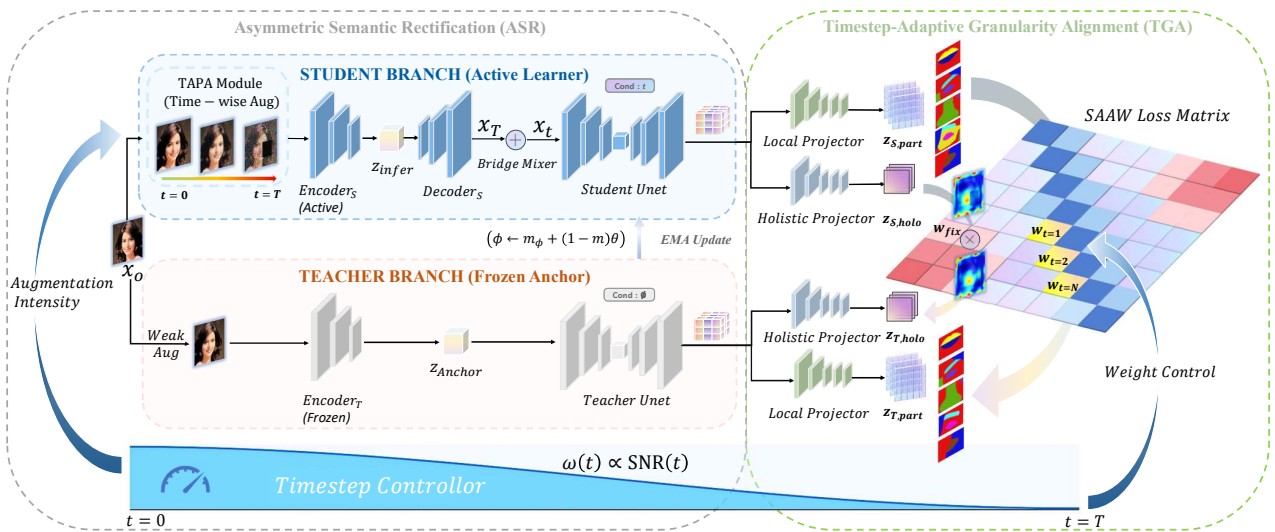

**Figure 2.** The ArcDAE Framework. We propose an asymmetric dual-branch architecture for unified representation learning. (Left) *asymmetric semantic rectification* (ASR): The framework establishes a *Clean-to-Noisy* consistency task. The teacher branch serves as a stable anchor ($\mathbf{x}_0$). In contrast, the student input is dynamically corrupted by *timestep-aware progressive augmentation* (TAPA), where noise and occlusion intensity scale with diffusion steps, forcing the encoder to infer semantic targets from increasingly degraded views. (Right) *timestep-adaptive granularity alignment* (TGA): The latent space is explicitly disentangled into translation-invariant global ($\mathbf{z}_{\text{holo}}$) and spatial-sensitive local ($\mathbf{z}_{\text{part}}$) granularities. Crucially, the alignment is modulated by *timestep-aware adaptive weighting* (TAAW), which utilizes the trajectory signal to dynamically balance the focus between semantic rectification and structural reconstruction.

Define the score-estimation residual as $\mathbf{e}_\theta(\mathbf{x}, t) := \mathbf{s}_\theta(\mathbf{x}, t) - \nabla_\mathbf{x} \log p_t(\mathbf{x})$, and the pointwise error as $\mathcal{E}(\theta; \mathbf{x}, t) = \|\mathbf{e}_\theta(\mathbf{x}, t)\|_2^2$. By the orthogonality of the projection, the *weighted Fisher divergence* $\mathcal{J}(\theta)$ admits an exact decomposition into semantic and nuisance components:

$$\mathcal{J}(\theta) = \int_0^T \lambda(t) \, \mathbb{E}_{p_t(\mathbf{x})} \left[ \|\mathbf{e}_\theta(\mathbf{x}, t)\|_2^2 \right] \mathrm{d}t$$
$$= \int_0^T \lambda(t) \, \mathbb{E}_{p_t(\mathbf{x})} \left[ \|\boldsymbol{\ell}_{\text{sem}}(\mathbf{x}, t)\|_2^2 + \|\boldsymbol{\ell}_{\text{nuis}}(\mathbf{x}, t)\|_2^2 \right] \mathrm{d}t. \tag{3}$$

where $\lambda(t) \in \mathbb{R}^+$ is the standard weighting function (e.g., $g(t)^2$), and the tangential (semantic) and normal (nuisance) error components are defined respectively as:

$$\begin{cases} \boldsymbol{\ell}_{\text{sem}}(\mathbf{x}, t) := \mathbf{\Pi}_\mathbf{x} \mathbf{e}_\theta(\mathbf{x}, t), \\ \boldsymbol{\ell}_{\text{nuis}}(\mathbf{x}, t) := (\mathbf{I}_D - \mathbf{\Pi}_\mathbf{x}) \mathbf{e}_\theta(\mathbf{x}, t). \end{cases} \tag{4}$$

**The Curse of Dimensionality in Ambient Space.** The dominance of $\mathcal{L}_{\text{nuis}}$ arises from concentration of measure in high-dimensional spaces (Ledoux, 2001). Consider a simplified setting where an ambient isotropic perturbation $\boldsymbol{\xi} \sim \mathcal{N}(\mathbf{0}, \mathbf{I}_D)$ is decomposed into tangent and normal components. Its squared projections onto the tangent space (dimension $d_\mathcal{M}$) and normal space (dimension $D - d_\mathcal{M}$) are denoted as $K := \|\mathbf{\Pi}_\mathbf{x} \boldsymbol{\xi}\|_2^2$ and $N := \|(\mathbf{I}_D - \mathbf{\Pi}_\mathbf{x}) \boldsymbol{\xi}\|_2^2$. These variables follow independent Chi-squared distributions: $K \sim \chi_{d_\mathcal{M}}^2$ and $N \sim \chi_{D-d_\mathcal{M}}^2$. Crucially, for $d_\mathcal{M} > 2$, the inverse moment is $\mathbb{E}[K^{-1}] = (d_\mathcal{M} - 2)^{-1}$. By indepen-

dence, the expected energy ratio factorizes exactly:

$$\mathbb{E}\left[ \frac{\|(\mathbf{I}_D - \mathbf{\Pi}_\mathbf{x})\boldsymbol{\xi}\|_2^2}{\|\mathbf{\Pi}_\mathbf{x}\boldsymbol{\xi}\|_2^2} \right] = \frac{D - d_\mathcal{M}}{d_\mathcal{M} - 2}$$
$$= \underbrace{\frac{D - d_\mathcal{M}}{d_\mathcal{M}} + \mathcal{O}\left(\frac{D}{d_\mathcal{M}^2}\right)}_{\text{Taylor expansion}}. \tag{5}$$

When $D \gg d_\mathcal{M}$, this ratio becomes large, indicating that score matching assigns disproportionate gradient energy to normal-space variation. This geometric imbalance motivates our *information overload* hypothesis: without additional semantic constraints, diffusion bridges may allocate excessive capacity to high-frequency nuisance factors rather than manifold-level semantics. Appendix A.2 provides the complete derivation.

This geometric imbalance leads to a fundamental bottleneck in diffusion-based representation learning, which we formally characterize as follows:

**Theorem 1** (Information Overload Trap)**.** *In a diffusion bridge* $(\mathbf{X}_t)_{t \in [0,T]}$, *if the intrinsic manifold dimension* $d_\mathcal{M} \ll D$ *and the latent bottleneck dimension* $d_z \ll D$, *the score-matching objective admits a high-noise regime in which normal-space components dominate the gradient energy. Consequently, a low-dimensional latent bottleneck must allocate capacity to nuisance variation unless additional semantic constraints are imposed. A formal statement and proof are provided in Appendix A.3.*

## 3.2. Asymmetric Semantic Rectification (ASR)

To mitigate information overload, ASR separates stable semantic factors from timestep-dependent stochastic variation. The teacher branch provides a clean semantic target, whereas the student branch learns to suppress nuisance variation that becomes dominant along noisy diffusion states.

**Definition 1** (Bifurcated Dynamics). Let $\mathbf{x}_0$ be the clean data. We define the teacher ($\mathcal{T}$) and student ($\mathcal{S}$) branches via the following coupled system:

$$
\begin{cases}
\mathbf{z}^{\mathcal{T}} = \mathrm{sg}\left[E_\phi(\mathbf{v}^{\mathcal{T}})\right], & \mathbf{v}^{\mathcal{T}} \sim p_{\mathrm{aug}}(\cdot \mid \mathbf{x}_0, 0), \\
\mathbf{x}_t^{\mathcal{S}} = \alpha_t \mathbf{v}^{\mathcal{S}} + \sigma_t \boldsymbol{\epsilon}, & \mathbf{v}^{\mathcal{S}} \sim p_{\mathrm{aug}}(\cdot \mid \mathbf{x}_0, \gamma(t)),
\end{cases}
\tag{6}
$$

where $\boldsymbol{\epsilon} \sim \mathcal{N}(\mathbf{0}, \mathbf{I}_D)$ denotes standard Gaussian noise, and $\mathrm{sg}[\cdot]$ indicates the stop-gradient operation. The teacher parameter $\phi$ is maintained as an *exponential moving average* (EMA) of the student parameter $\theta$, $p_{\mathrm{aug}}(\cdot \mid \mathbf{x}, \gamma)$ represents the data augmentation distribution with intensity $\gamma$.

The ASR objective, $\mathcal{L}_{\mathrm{ASR}} \coloneqq \mathbb{E}\left[\|\mathbf{z}^{\mathcal{T}} - E_\theta(\mathbf{x}_t^{\mathcal{S}})\|_2^2\right]$, implicitly imposes a geometric low-pass filtering effect on the underlying semantic manifold. For compact notation, we write $E \equiv E_\theta$ when no ambiguity arises.

**Theorem 2** (Manifold Smoothness via Jacobian Regularization). *Let $E_\theta : \mathbb{R}^D \to \mathbb{R}^{d_h}$ be a $C^3$ smooth mapping and let $\mathbf{m}_t \coloneqq \alpha_t \mathbf{v}^{\mathcal{S}}$. In the locally aligned semantic regime where $\|E_\theta(\mathbf{m}_t) - \mathbf{z}^{\mathcal{T}}\|_2$ is sufficiently small, the ASR objective admits the following leading-order expansion with an explicit residual–Hessian remainder:*

$$
\mathcal{L}_{\mathrm{ASR}} = \underbrace{\left\|E_\theta(\mathbf{m}_t) - \mathbf{z}^{\mathcal{T}}\right\|_2^2}_{\text{Alignment error}} + \underbrace{\sigma_t^2\, \mathrm{Tr}\left(\mathbf{J}_{E_\theta}(\mathbf{m}_t)^\top \mathbf{J}_{E_\theta}(\mathbf{m}_t)\right)}_{\text{Jacobian regularization}}
$$
$$
+ \mathcal{O}\left(\sigma_t^2 \left\|E_\theta(\mathbf{m}_t) - \mathbf{z}^{\mathcal{T}}\right\|_2 \|\mathbf{H}_{E_\theta}(\mathbf{m}_t)\|_{\mathrm{op}} + \sigma_t^4\right).
\tag{7}
$$

The detailed derivation is provided in Appendix A.4.

**Contrastive Rectification.** Although normalized MSE can align teacher and student features, in the high-noise regime it may over-enforce the geometric contraction effect characterized by Theorem 2, driving the encoder toward locally constant representations. We instead adopt InfoNCE, whose alignment term preserves semantic consistency while its uniformity term maintains robust hyperspherical feature dispersion and mitigates dimensional collapse (Wang & Isola, 2020). Appendix A.5 provides further comparison and ablation evidence.

*Remark* 1 (Dynamic Regularization vs. Static Contractive AE). While Theorem 2 establishes a link to the *contractive autoencoder* (CAE), a critical distinction lies in the time-dependent variance $\sigma_t^2$. In classical CAE, the regularization weight is a static hyperparameter. In our *diffusion bridge*, $\sigma_t^2$ acts as an adaptive annealing schedule:

- As $t \to 0$ (near data), $\sigma_t^2 \to 0$, so the contraction weakens and high-frequency reconstruction details can be well preserved.
- As $t \to T$ (near noise), $\sigma_t^2$ increases the contraction strength, reducing the sensitivity to unstable local perturbations and biasing the representation toward stable global semantics.

This time-dependent contraction provides a principled way to balance semantic stability and reconstructive fidelity.

## 3.3. Timestep-Aware Progressive Augmentation (TAPA)

TAPA prevents the student encoder from relying on low-level shortcuts by increasing augmentation strength with the diffusion timestep $t \in [0, T]$. We define the progressive intensity as $\gamma(t) = t/T$. The strong augmentation operator is $\Psi_t = \psi_{\mathrm{jitter},t} \circ \psi_{\mathrm{blur},t} \circ \psi_{\mathrm{crop},t}$, which applies cropping, blurring, and color jittering in sequence. For a given input $\mathbf{x}_0$, the asymmetric view construction flow is defined as:

$$
\begin{cases}
\mathbf{v}^{\mathcal{T}} = \mathcal{A}_{\mathrm{weak}}(\mathbf{x}_0), & \text{(Teacher Branch)} \\
\mathbf{x}_t^{\mathcal{S}} = \sqrt{\bar{\alpha}_t}\Psi_t(\mathbf{x}_0) + \sqrt{1 - \bar{\alpha}_t}\boldsymbol{\epsilon}, & \text{(Student Branch)}
\end{cases}
\tag{8}
$$

where $\boldsymbol{\epsilon} \sim \mathcal{N}(\mathbf{0}, \mathbf{I}_D)$ denotes standard Gaussian noise. Here, $\bar{\alpha}_t = \prod_{i=1}^t (1 - \beta_i)$ represents the cumulative signal variance determined by the noise schedule $\beta_t$, and $\mathcal{A}_{\mathrm{weak}}$ denotes standard weak augmentations such as random flip. The sub-operators in $\Psi_t$ are explicitly parameterized by $\gamma(t)$ to implement an adaptive *information sifting* task:

- **Progressive Crop** ($\psi_{\mathrm{crop},t}$): We define a time-dependent scale range $\mathbb{S}_t = [s_t, 1.0]$. The student views are cropped with a scale $s \sim \mathcal{U}(\mathbb{S}_t)$, where the lower bound $s_t$ follows:

$$
s_t = 1.0 - (1.0 - s_{\min}) \cdot \gamma(t).
\tag{9}
$$

By setting $s_{\min} = 0.2$, the student receives approximately 20% of the original spatial evidence as $t \to T$, distributed across sparse blocks in our implementation. This retains sparse global layout cues while requiring the student to infer semantics from incomplete evidence under severe corruption.

- **Adaptive Gaussian Blur** ($\psi_{\mathrm{blur},t}$): To eliminate high-frequency identity leakage, we apply a Gaussian kernel with an adaptive standard deviation $\sigma_{\mathrm{blur}}(t)$:

$$
\sigma_{\mathrm{blur}}(t) = \sigma_{\mathrm{base}} + \Delta\sigma_{\mathrm{blur}} \cdot \gamma(t),
\tag{10}
$$

where $\sigma_{\mathrm{base}} = 0.1$ and $\Delta\sigma_{\mathrm{blur}} = 2.0$. This ensures the student focuses on coarse-grained semantic priors rather than local textures in noise-intensive regimes.

- **Progressive Color Jitter** ($\psi_{\mathrm{jitter},t}$): The intensity of brightness, contrast, and saturation $\{\kappa_i\}_{i=1}^3$ is modulated via:

$$
\kappa_i \in [1 - \eta_{\mathrm{jitter}} \cdot \gamma(t), 1 + \eta_{\mathrm{jitter}} \cdot \gamma(t)],
\tag{11}
$$

where $\eta_{\text{jitter}}$ (e.g., $\eta_{\text{jitter}} = 0.4$) is the maximum jitter factor. This discourages the representation from relying on low-level color statistics.

### 3.4. Timestep-Aware Adaptive Weighting (TAAW)

To govern the information flow between global semantics and local structures during the diffusion process, TAAW utilizes the *signal-to-noise ratio* (SNR) of the current bridge state to gate the alignment objective. Under the VP schedule, we define $\text{SNR}(t) := \bar{\alpha}_t / (1 - \bar{\alpha}_t)$, where $\bar{\alpha}_t$ is the cumulative variance-preserving coefficient.

We define the *signal-aware gating* function $\omega(t)$ as a monotonic mapping derived from the diffusion schedule:

$$\omega(t) \equiv \omega_{\tau_{\text{snr}}}(t) := \frac{\text{SNR}(t)^{1/\tau_{\text{snr}}}}{1 + \text{SNR}(t)^{1/\tau_{\text{snr}}}}, \qquad \tau_{\text{snr}} > 0. \tag{12}$$

When $\tau_{\text{snr}} = 1$, this recovers $\omega(t) = \text{SNR}(t)/(1 + \text{SNR}(t)) = \bar{\alpha}_t$ under the VP schedule.

Here, $\mathbf{z}_{\text{holo}}$ and $\mathbf{z}_{\text{part}}$ denote the outputs of the global and local projection heads, respectively. Under this formulation:

1. When $t \to 0$ (high SNR), $\omega(t) \to 1$, compelling the student to maintain high-fidelity structural correspondence for precise reconstruction.

2. As $t \to T$ (low SNR), $\omega(t) \to 0$, attenuating local alignment when the observation is noise dominated. This reduces the incentive to encode high-entropy nuisance variation in the bottleneck.

Let the feature map be $\mathbf{h} : \Omega \to \mathbb{R}^C$, where $\Omega$ denotes the spatial domain, and let $\mathcal{P}_{\text{holo}}, \mathcal{P}_{\text{part}} : \mathbb{R}^C \to \mathbb{R}^{d_h}$ denote the holistic and partial projection heads. We formulate both projectors as functional operators over $\Omega$:

**Holistic Projector.** This treats global context as an integration over the uniform measure $\mu$ on $\Omega$:

$$\mathbf{z}_{\text{holo}} = \mathcal{P}_{\text{holo}} \left( \int_\Omega \mathbf{h}(\mathbf{u}) \, d\mu(\mathbf{u}) \right). \tag{13}$$

**Partial Projector.** Models local saliency via a Gibbs measure $\rho_{\mathbf{q}}$. With a learnable query $\mathbf{q} \in \mathbb{R}^{d_k}$ and key projection $\phi_K : \mathbb{R}^C \to \mathbb{R}^{d_k}$, the partition function and Radon-Nikodym derivative are:

$$Z_\rho(\mathbf{q}) := \int_\Omega \exp\left( \frac{\langle \mathbf{q}, \phi_K(\mathbf{h}(\mathbf{u})) \rangle}{\sqrt{d_k}} \right) d\mu(\mathbf{u}),$$
$$\frac{d\rho_{\mathbf{q}}}{d\mu}(\mathbf{u}) = \frac{1}{Z_\rho(\mathbf{q})} \exp\left( \frac{\langle \mathbf{q}, \phi_K(\mathbf{h}(\mathbf{u})) \rangle}{\sqrt{d_k}} \right). \tag{14}$$

The partial representation is then the expectation under:

$$\mathbf{z}_{\text{part}} = \mathcal{P}_{\text{part}} \left( \int_\Omega \mathbf{h}(\mathbf{u}) \, d\rho_{\mathbf{q}}(\mathbf{u}) \right). \tag{15}$$

---

**Algorithm 1** ArcDAE Training Procedure

**Input:** Dataset $\mathcal{D}$, Student Network $\mathcal{S}_\theta$, Teacher Network $\mathcal{T}_\phi$, Diffusion Steps $T$, Training Steps $N_{\text{train}}$.
**Hyperparams:** Learning rate $\alpha_{\text{lr}}$, EMA decay $m$, Balance weight $\lambda_{\text{align}}$.
**Initialize:** $\phi \leftarrow \theta$.
**Output:** Optimized Student Model $\mathcal{S}_\theta$.
**for** training step $k = 1$ to $N_{\text{train}}$ **do**
   *1. TAPA View Construction:*
   Sample batch $\mathbf{x} \sim \mathcal{D}$ and timestep $t \sim \mathcal{U}(1, T)$.
   Sample noise $\boldsymbol{\epsilon} \sim \mathcal{N}(\mathbf{0}, \mathbf{I}_D)$.
   Set dynamic intensity: $\gamma(t) \leftarrow t/T$.
   $\mathbf{x}_{\mathcal{T}} \leftarrow \mathcal{A}_{\text{weak}}(\mathbf{x})$        *{Semantic Anchor}*
   $\tilde{\mathbf{x}}_{\mathcal{S}} \leftarrow \Psi_t(\mathbf{x})$      *{Structurally Degraded Input}*
   *2. Asymmetric Forward Process:*
   $\tilde{\mathbf{x}}_t \leftarrow \sqrt{\bar{\alpha}_t} \tilde{\mathbf{x}}_{\mathcal{S}} + \sqrt{1 - \bar{\alpha}_t} \boldsymbol{\epsilon}$   *{Diffusion Corruption}*
   *3. Semantic Rectification & Inference:*
   $\mathbf{z}_{\mathcal{T}}^{\text{holo}}, \mathbf{z}_{\mathcal{T}}^{\text{part}} \leftarrow \mathcal{T}_\phi(\mathbf{x}_{\mathcal{T}})$    *{Teacher: Clean Semantic Extraction}*
   $\mathbf{z}_{\mathcal{S}}^{\text{holo}}, \mathbf{z}_{\mathcal{S}}^{\text{part}}, \hat{\boldsymbol{\epsilon}} \leftarrow \mathcal{S}_\theta(\tilde{\mathbf{x}}_t, t)$    *{Student: Denoising & Prediction}*
   *4. Synergistic Optimization (via TGA):*
   Calculate adaptive weight: $\omega \leftarrow \omega_{\tau_{\text{snr}}}(t)$.
   $\mathcal{L}_{\text{TGA}} \leftarrow \ell_{\text{NCE}}(\mathbf{z}_{\mathcal{S}}^{\text{holo}}, \mathbf{z}_{\mathcal{T}}^{\text{holo}}) + \omega \cdot \ell_{\text{NCE}}(\mathbf{z}_{\mathcal{S}}^{\text{part}}, \mathbf{z}_{\mathcal{T}}^{\text{part}})$
   $\mathcal{L}_{\text{diff}} \leftarrow \| \boldsymbol{\epsilon} - \hat{\boldsymbol{\epsilon}} \|_2^2$
   $\mathcal{L}_{\text{total}} \leftarrow \mathcal{L}_{\text{diff}} + \lambda_{\text{align}} \cdot \mathcal{L}_{\text{TGA}}$
   *5. Parameter Update:*
   $\theta \leftarrow \theta - \alpha_{\text{lr}} \nabla_\theta \mathcal{L}_{\text{total}}$
   $\phi \leftarrow m\phi + (1 - m)\theta$       *{EMA Update}*
**end for**

---

**Resolution Invariance.** Formulating projectors as integral operators confers *discretization invariance*. Unlike discrete attention, this ensures $\mathbf{z}_{\text{holo}}$ and $\mathbf{z}_{\text{part}}$ converge to fixed points independent of input resolution, enabling generalization across varying granularities without recalibration.

**Optimization Objective.** We maximize the mutual information lower bound between the teacher and student trajectories. Let $\lambda_{\text{align}} > 0$ balance the alignment strength and $\mathcal{L}_{\text{diff}}(\theta) := \mathcal{J}(\theta)$ be the diffusion loss from Eq. (3). The total objective is:

$$\mathcal{L}_{\text{total}} = \mathcal{L}_{\text{diff}}(\theta)$$
$$+ \lambda_{\text{align}} \left[ \ell_{\text{NCE}}(\mathbf{z}_{\text{holo}}^{\mathcal{S}}, \mathbf{z}_{\text{holo}}^{\mathcal{T}}) + \omega(t) \cdot \ell_{\text{NCE}}(\mathbf{z}_{\text{part}}^{\mathcal{S}}, \mathbf{z}_{\text{part}}^{\mathcal{T}}) \right]. \tag{16}$$

**Resolution of Information Overload.** While $\mathcal{L}_{\text{diff}}$ compels the encoding of high-frequency nuisance variables, our gated alignment acts as a *semantic sieve*. By driving $\omega(t) \to 0$ in noise-dominated regimes, we decouple the latent representation from the entropic noise, ensuring convergence exclusively onto the clean data manifold.

*Table 1.* Linear-probe attribute prediction quality comparison. Models are trained on CelebA and FFHQ with dim(**z**)=$d_z = 512$. *Gen* indicates generation capability. Bold indicates the best result, and underline indicates the second-best result.

| METHOD | GEN | CELEBA | | | FFHQ | | |
|---|---|---|---|---|---|---|---|
| | | AP (↑) | PEARSON'S $r$ (↑) | MSE (↓) | AP (↑) | PEARSON'S $r$ (↑) | MSE (↓) |
| SIMCLR (CHEN ET AL., 2020) | ✗ | 0.597 | 0.474 | 0.603 | 0.608 | 0.481 | 0.638 |
| $\beta$-TCVAE (CHEN ET AL., 2018) | ✓ | 0.450 | 0.378 | 0.573 | 0.432 | 0.335 | 0.608 |
| IB-GAN (JEON ET AL., 2021) | ✓ | 0.442 | 0.307 | 0.597 | 0.428 | 0.260 | 0.644 |
| DIFFAE (PREECHAKUL ET AL., 2022) | ✓ | 0.603 | 0.598 | 0.421 | 0.605 | 0.606 | 0.410 |
| PDAE (ZHANG ET AL., 2022) | ✓ | 0.602 | 0.596 | 0.410 | 0.597 | 0.603 | 0.416 |
| DITI (YUE ET AL., 2024) | ✓ | 0.623 | 0.617 | 0.392 | 0.614 | 0.622 | 0.384 |
| DBAE (DET.) (KIM ET AL., 2025) | ✓ | 0.650 | 0.635 | 0.413 | 0.656 | 0.638 | 0.404 |
| DBAE (STOCH.) (KIM ET AL., 2025) | ✓ | 0.655 | 0.643 | 0.369 | 0.664 | 0.675 | 0.332 |
| ARCDAE (OURS) | ✓ | **0.676** | **0.656** | **0.345** | **0.681** | **0.718** | **0.319** |

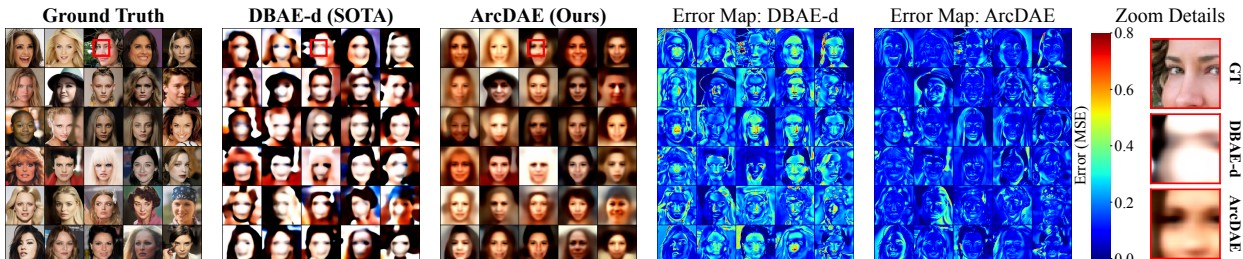

*Figure 3.* Early Training Convergence Analysis. Visual comparison of reconstruction quality on FFHQ at an early training stage (20, 000 steps). **Reconstruction (Left):** Compared to the Ground Truth, the SOTA baseline DBAE-d (Kim et al., 2025) exhibits severe structural collapse and high-frequency artifacts. In contrast, ArcDAE successfully reconstructs coherent facial structures and preserves identity details. **Error Analysis (Right):** The L1 Error Maps (cols 4-5) visualize pixel-level deviation, where ArcDAE shows significantly lower error magnitude (darker blue) compared to the baseline. The **Zoom Details** (far right) further confirm that our adaptive noise sifting strategy efficiently captures high-level features and eliminates artifacts even with limited training iterations.

*Table 2.* Autoencoding reconstruction quality comparison among tractable and 512-dimensional latent variable models. Bold indicates the best result, and underline indicates the second-best result.

| METHOD | TRACTABILITY | LATENT DIM (↓) | SSIM (↑) | LPIPS (↓) | MSE (↓) |
|---|---|---|---|---|---|
| STYLEGAN2 ($\mathcal{W}$) (KARRAS ET AL., 2020) | ✗ | 512 | 0.677 | 0.168 | 0.016 |
| STYLEGAN2 ($\mathcal{W}+$) (ABDAL ET AL., 2019) | ✗ | 7,168 | 0.827 | 0.114 | 0.006 |
| VQ-GAN (ESSER ET AL., 2021) | ✓ | 65,536 | 0.782 | 0.109 | $3.61 \times 10^{-3}$ |
| VQ-VAE2 (RAZAVI ET AL., 2019) | ✓ | 327,680 | 0.947 | 0.012 | $4.87 \times 10^{-4}$ |
| NVAE (VAHDAT & KAUTZ, 2020) | ✓ | 6,005,760 | 0.984 | 0.001 | $4.85 \times 10^{-5}$ |
| DDIM (INFERRED $\mathbf{x}_T$) (SONG ET AL., 2021A) | ✗ | 49,152 | 0.917 | 0.063 | 0.002 |
| DIFFAE (INFERRED $\mathbf{x}_T$) (PREECHAKUL ET AL., 2022) | ✗ | 49,664 | 0.991 | 0.011 | $6.07 \times 10^{-5}$ |
| PDAE (INFERRED $\mathbf{x}_T$) (ZHANG ET AL., 2022) | ✗ | 49,664 | 0.994 | 0.007 | $3.84 \times 10^{-5}$ |
| DIFFAE (RANDOM $\mathbf{x}_T$) (PREECHAKUL ET AL., 2022) | ✓ | 512 | 0.677 | 0.073 | 0.007 |
| PDAE (RANDOM $\mathbf{x}_T$) (ZHANG ET AL., 2022) | ✓ | 512 | 0.689 | 0.098 | $5.01 \times 10^{-3}$ |
| DBAE (STOCHASTIC) (KIM ET AL., 2025) | ✓ | 512 | 0.920 | 0.094 | $4.81 \times 10^{-3}$ |
| DBAE-D (DETERMINISTIC) (KIM ET AL., 2025) | ✓ | 512 | 0.953 | 0.072 | $2.49 \times 10^{-3}$ |
| ARCDAE (OURS) | ✓ | 512 | **0.962** | **0.065** | $2.79 \times 10^{-3}$ |

## 4. Experiments

We evaluate ArcDAE on CelebA (Liu et al., 2015) and FFHQ (Karras et al., 2019) against state-of-the-art diffusion representation benchmarks.

**Implementation Details.** We adopt the ADM U-Net architecture for both the score network $\epsilon_\theta$ and encoder $E_\phi$ with a latent dimension of $d_z = 512$, following DBAE (Kim et al., 2025). Training utilizes 4 NVIDIA RTX 4090 GPUs. TAPA scales linearly with $t$, and TAAW follows the variance-preserving SNR schedule. For generation, a standard *latent diffusion model* (LDM) is trained as the prior

$p_\omega(\mathbf{z})$. Detailed experimental configurations and implementation Details are provided in Appendix B.

### 4.1. Semantic Density Evaluation via Downstream Inference

We perform linear probing to quantify the semantic density of the latent space **z**. A higher accuracy indicates better disentanglement of semantics from noise. **Results:** Table 1 reports the performance. ArcDAE establishes a new state-of-the-art across all metrics. Notably, on FFHQ, ArcDAE achieves **0.681 AP**, surpassing both the deterministic DBAE (0.656) and its stochastic variant (0.664). This confirms that

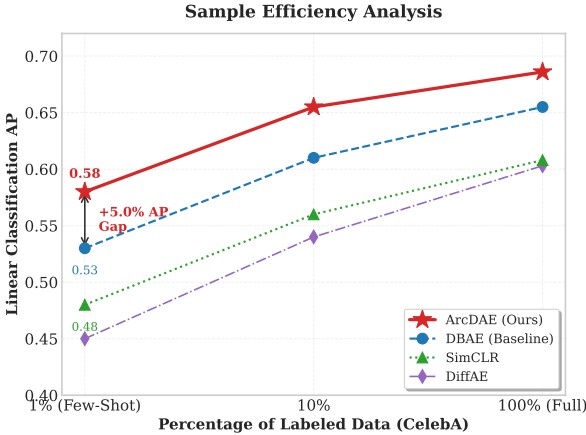

*Figure 4.* Sample Efficiency on CelebA. Linear probe accuracy across varying label ratios (1%, 10%, 100%). ArcDAE (red star) demonstrates superior robustness, achieving **0.58 AP** with only 1% labels, significantly outperforming other models.

*Table 3.* Disentanglement and sample quality on CelebA. ArcDAE achieves better disentanglement (TAD) without the image quality degradation seen in baselines. Bold indicates the best result, and underline indicates the second-best result.

| METHOD | REG | TAD (↑) | ATTRS (↑) | FID (↓) |
|---|---|---|---|---|
| AE | ✗ | 0.042±.004 | 1.0±.0 | 90.4±1.8 |
| DIFFAE (PREECHAKUL ET AL., 2022) | ✗ | 0.155±.010 | 2.0±.0 | 22.7±2.1 |
| DBAE (KIM ET AL., 2025) | ✗ | 0.124±.078 | 2.2±1.3 | 11.8±.2 |
| ARCDAE | ✗ | **0.210**±.070 | **3.9**±.6 | **10.28**±.2 |
| VAE (KINGMA & WELLING, 2014) | ✓ | 0.000±.000 | 0.0±.0 | 94.3±2.8 |
| β-VAE (HIGGINS ET AL., 2017) | ✓ | 0.088±.051 | 1.6±.8 | 99.8±2.4 |
| INFOVAE (ZHAO ET AL., 2019) | ✓ | 0.000±.000 | 0.0±.0 | 77.8±1.6 |
| INFODIFF (WANG ET AL., 2023) | ✓ | 0.299±.006 | 3.0±.0 | 22.3±1.2 |
| DISDIFF (YANG ET AL., 2023) | ✓ | 0.305±.010 | – | 18.3±2.1 |
| DBAE+TC (KIM ET AL., 2025) | ✓ | 0.362±.036 | 3.8±.8 | 13.4±.2 |
| ARCDAE+TC | ✓ | **0.385**±.036 | **4.9**±.8 | **12.2**±.2 |

*Table 4.* Unconditional generation results on FFHQ. Bold indicates the best result, and underline indicates the second-best result.

| METHOD | PREC (↑) | IS (↑) | FID (↓) | REC (↑) |
|---|---|---|---|---|
| DDPM (HO ET AL., 2020) | 0.768 | 3.11 | **9.14** | 0.335 |
| DIFFAE (PREECHAKUL ET AL., 2022) | 0.762 | 2.98 | 9.40 | **0.458** |
| PDAE (ZHANG ET AL., 2022) | 0.695 | 2.23 | 47.42 | 0.153 |
| DBAE (KIM ET AL., 2025) | 0.780 | 3.87 | 11.25 | 0.392 |
| ARCDAE (OURS) | **0.785** | **3.92** | 10.28 | 0.437 |
| DIFFAE + AE | 0.750 | 3.63 | 2.84 | 0.685 |
| DBAE + AE | 0.751 | 3.57 | 1.77 | 0.687 |
| ARCDAE + AE | **0.792** | **4.10** | **1.65** | **0.710** |

active noise sifting (ASR) yields a more linearly separable semantic manifold than the passive reconstruction objectives

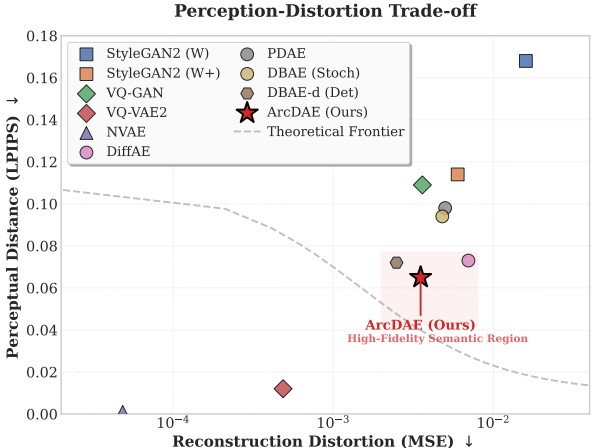

*Figure 5.* Perception-Distortion Trade-off Analysis. Comparison on CelebA-HQ. Lower is better for both axes. ArcDAE sacrifices negligible pixel-alignment (MSE) for superior perceptual quality (LPIPS), effectively resolving the information overload problem.

used in baselines.

### 4.2. Generative Dynamics, Disentanglement and Quality

**Training Dynamics.** Figure 3 illustrates training stability at 20k iterations. Unlike DBAE-d, which suffers from structural collapse and color saturation due to noise overfitting, ArcDAE maintains structural coherence. This validates that TAPA and TAAW effectively guide the model to learn semantic manifolds rather than memorizing noise patterns.

**Disentanglement and Unconditional Generation.** We further investigate the latent structure in Table 3. Under Total Correlation (TC) regularization, ArcDAE+TC establishes a new Pareto frontier, improving disentanglement (TAD **0.385**) while maintaining superior sample quality (FID **12.2**) compared to DBAE+TC (TAD 0.362, FID 13.4).

**Unconditional Generation.** We train a latent prior $p_\omega(\mathbf{z})$ to sample novel images. Table 4 shows that ArcDAE+AE achieves the best FID (**1.65**), while ArcDAE without the auxiliary AE prior achieves the best precision among non-AE priors on FFHQ. The *sifted* latent manifold of ArcDAE is smoother than the noise-entangled manifold of DBAE, facilitating more accurate prior modeling and resulting in superior diversity for ArcDAE+AE (Recall **0.710**). **Sample Efficiency.** We conduct few shot classification on CelebA to demonstrate semantic feature learning. Figure 4 illustrates that ArcDAE achieves an AP of **0.58** with only 1% of labels and outperforms DBAE (0.53). This confirms ASR rectified features are robust and data efficient.

### 4.3. Perceptual Fidelity vs. Noise Fitting

We evaluate reconstruction quality to assess information completeness. Table 2 compares tractable latent models on

CelebA-HQ (Karras et al., 2018).

**Perception-Distortion Trade-off:** As predicted by Theorem 1, minimizing MSE in diffusion bridges often leads to noise overfitting. While DBAE-d achieves the lowest MSE $(2.49 \times 10^{-3})$ by memorizing high-frequency noise, ArcDAE prioritizes structural semantics, resulting in a slightly higher MSE $(2.79 \times 10^{-3})$ but significantly better perceptual quality (LPIPS **0.065** vs. 0.072). Figure 5 visualizes this trade-off: ArcDAE occupies a high-fidelity region distinct from the noise-dominated regime of DBAE.

### 4.4. Ablation Studies and Sensitivity Analysis

**Teacher Necessity.** Removing the teacher anchor from TAPA causes collapse (AP 0.125), because high-noise timesteps leave the student with insufficient semantic evidence and bias the encoder toward a low-rank representation; Appendix C.3 reports the corresponding SVD entropy diagnostics.

**Dual Granularity.** Removing TGA increases LPIPS by 0.02, confirming that holistic and partial alignment are complementary. **Sensitivity.** Table 6 shows that ArcDAE is stable for $\lambda_{align} \in [0.01, 0.1]$, with FFHQ peaking at 0.1 and CelebA peaking at 0.01.

*Table 5.* Component ablation on FFHQ. Bold indicates the best result, and underline indicates the second-best result.

| VARIANT | AP (LIN) | LPIPS (REC) | FID (GEN) |
|---|---|---|---|
| BASELINE (DBAE) | 0.664 | 0.075 | 11.25 |
| + ASR (STATIC WEAK) | 0.666 | 0.072 | 11.10 |
| + TAPA (NO TEACHER) | 0.125 | 0.240 | 45.30 |
| + ASR (DYN. TAPA) | 0.679 | 0.068 | 10.72 |
| + ASR + TGA (W/O TAAW) | 0.675 | 0.066 | 10.85 |
| FULL ARCDAE | **0.681** | **0.065** | **10.28** |

*Table 6.* Sensitivity of $\lambda_{align}$. Bold indicates the best result, and underline indicates the second-best result.

| DATASET | METRIC | $\lambda_{align} = 0.01$ | $\lambda_{align} = 0.1$ | $\lambda_{align} = 1.0$ |
|---|---|---|---|---|
| FFHQ | AP (↑) | 0.665 | **0.681** | 0.620 |
| (128×128) | MSE (↓) | 0.330 | **0.319** | 0.345 |
| CELEBA | AP (↑) | **0.676** | 0.642 | 0.547 |
| (64×64) | MSE (↓) | **0.345** | 0.372 | 0.402 |

## 5. Conclusion

In this work, we presented ArcDAE, which redefines the diffusion bridge as an engine of *asymmetric information sifting* rather than a passive *information overload carrier*. We remove high-frequency redundancy while strictly maintaining the coupled mapping to avoid an *information split*. Specifically, we introduced *asymmetric semantic rectification* to orthogonalize semantic manifolds from stochastic noise distributions by leveraging a stable teacher anchor. This is complemented by *timestep-aware adaptive weighting*, which dynamically balances structural integrity and abstract semantics across the diffusion trajectory. ArcDAE

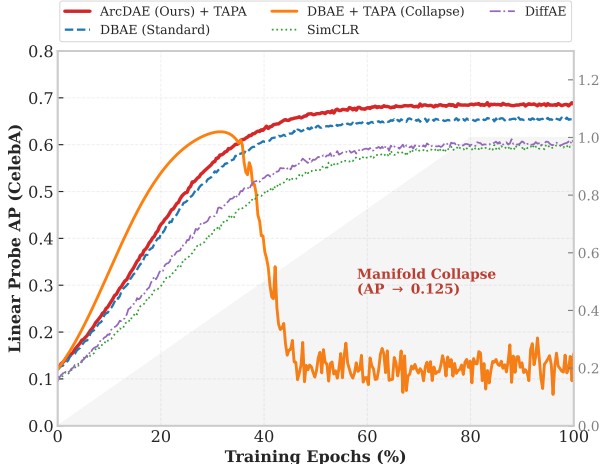

*Figure 6.* Training dynamics and manifold collapse. We plot linear-probe AP under the standard TAPA protocol with randomly sampled timestep $t$. The DBAE with TAPA (without teacher) variant collapses under extreme-noise samples, whereas ArcDAE remains stable due to the asymmetric teacher anchor.

explicitly treats the bridge as a dynamic sifter for noise-rectified representation learning. We further provided theoretical analyses characterizing the dimensional imbalance underlying information overload. Extensive experiments on CelebA, FFHQ, and additional non-face benchmarks demonstrated that ArcDAE consistently acquires more discriminative representations and superior perceptual fidelity than prior diffusion autoencoding methods. These results suggest that the proposed rectification mechanism generalizes beyond face-centric distributions and provides a scalable path toward unified models for simultaneous synthesis and understanding. Future research will focus on tighter theoretical bounds on rectification dynamics and extensions toward multimodal and large-scale representation learning.

## Acknowledgments

Shuo Chen was supported by National Major S&T Special Project on New Generation Artificial Intelligence (Nos. 2025ZD0123500), NSFC (Nos. 62506155), Provincial Natural Science Fund of Jiangsu (Nos. BK20251985), and Suzhou Municipal Leading Talents Fund (Nos. ZXL2025320). Zeke Xie was supported by Guangdong Provincial Key Lab of Integrated Communication, Sensing and Computation for Ubiquitous Internet of Things (No. 2023B1212010007). Jian Yang was supported by the NSFC under Grant Nos. U24A20330 and 62361166670.

## Impact Statement

This paper presents work whose goal is to advance the field of Machine Learning. There are many potential societal consequences of our work, none which we feel must be specifically highlighted here.

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

# A. Theoretical Analysis and Proofs

In this section, we provide rigorous derivations for the theorems presented in the main text.

## A.1. Filtration-Adapted Bridge SDE via Doob's $h$-transform

The drift term in Eq. (1) explicitly contains the terminal boundary $\mathbf{x}_T$, which may appear to depend on a future realization. In ArcDAE, however, we first sample the latent $\mathbf{z}$ and set $\mathbf{x}_T = \mathrm{Dec}(\mathbf{z})$ before simulating the bridge. Conditional on this sampled endpoint, $\mathbf{x}_T$ is fixed throughout the trajectory and can be treated as an $\mathcal{F}_0$-measurable boundary parameter under the enlarged filtration $\widetilde{\mathcal{F}}_t = \mathcal{F}_t \vee \sigma(\mathbf{x}_T)$. It is therefore not a random variable revealed in the future along the same trajectory.

Formally, the bridge can be constructed from an unconditioned reference process using Doob's $h$-transform. The transformed drift contains the logarithmic gradient of the transition kernel,

$$\mathbf{s}^*(\mathbf{X}_t, t; \mathbf{x}_T) = \nabla_{\mathbf{X}_t} \log p_{T|t}(\mathbf{x}_T \mid \mathbf{X}_t).$$

Since $\mathbf{X}_t$ is $\mathcal{F}_t$-adapted and $\mathbf{x}_T$ is $\sigma(\mathbf{x}_T)$-measurable at time zero under the enlarged filtration, the mapping $\mathbf{X}_t \mapsto \nabla_{\mathbf{X}_t} \log p_{T|t}(\mathbf{x}_T \mid \mathbf{X}_t)$ is $\widetilde{\mathcal{F}}_t$-measurable. Consequently, the modified drift is non-anticipating with respect to $\{\widetilde{\mathcal{F}}_t\}_{t \geq 0}$ and Eq. (1) defines a valid Itô SDE.

## A.2. Derivation of the Expected Normal-to-Tangent Energy Ratio

Let $\boldsymbol{\xi} \sim \mathcal{N}(\mathbf{0}, \mathbf{I}_D)$ be an isotropic perturbation in the ambient space $\mathbb{R}^D$. For a point $\mathbf{x}$ on a $d_{\mathcal{M}}$-dimensional differentiable manifold $\mathcal{M}$, let $\boldsymbol{\Pi}_{\mathbf{x}}$ denote the orthogonal projection onto the tangent space $\mathcal{T}_{\mathbf{x}}\mathcal{M}$. Define

$$K := \|\boldsymbol{\Pi}_{\mathbf{x}}\boldsymbol{\xi}\|_2^2, \qquad N := \|(\mathbf{I}_D - \boldsymbol{\Pi}_{\mathbf{x}})\boldsymbol{\xi}\|_2^2.$$

By isotropy and orthogonality, $K \sim \chi^2_{d_{\mathcal{M}}}$ and $N \sim \chi^2_{D-d_{\mathcal{M}}}$. Conditioned on $\mathbf{x}$, the projection matrix $\boldsymbol{\Pi}_{\mathbf{x}}$ is deterministic because it is determined only by the local manifold geometry at $\mathbf{x}$ and is independent of the isotropic perturbation $\boldsymbol{\xi}$. Equivalently, in an orthonormal basis that diagonalizes $\boldsymbol{\Pi}_{\mathbf{x}}$, the tangent and normal components of $\boldsymbol{\xi}$ occupy disjoint coordinates of a standard Gaussian vector; hence $\boldsymbol{\Pi}_{\mathbf{x}}\boldsymbol{\xi}$ and $(\mathbf{I}_D - \boldsymbol{\Pi}_{\mathbf{x}})\boldsymbol{\xi}$ are independent, and so $K$ and $N$ are strictly independent. For $d_{\mathcal{M}} > 2$, $K^{-1}$ is integrable and

$$\mathbb{E}[K^{-1}] = \frac{1}{d_{\mathcal{M}} - 2}, \qquad \mathbb{E}[N] = D - d_{\mathcal{M}}.$$

Therefore,

$$\mathbb{E}\left[\frac{N}{K}\right] = \mathbb{E}[N]\mathbb{E}[K^{-1}] \quad \text{by the independence of } N \text{ and } K = \frac{D - d_{\mathcal{M}}}{d_{\mathcal{M}} - 2}.$$

Equivalently,

$$\frac{D - d_{\mathcal{M}}}{d_{\mathcal{M}} - 2} = \frac{D - d_{\mathcal{M}}}{d_{\mathcal{M}}} \left(1 - \frac{2}{d_{\mathcal{M}}}\right)^{-1} = \frac{D - d_{\mathcal{M}}}{d_{\mathcal{M}}} \left(1 + \frac{2}{d_{\mathcal{M}}} + \mathcal{O}(d_{\mathcal{M}}^{-2})\right),$$

where the last equality follows from the Taylor expansion of $(1 - x)^{-1}$. This derivation formalizes the normal-space energy dominance when $D \gg d_{\mathcal{M}}$, explaining why unconstrained bridge objectives tend to allocate capacity to high-frequency nuisance variation.

## A.3. Proof of Theorem 1 (Rigorous Mathematical Formulation of Information Overload in Diffusion Bridges)

Let $(\Omega, \mathcal{F}, \mathbb{P})$ be a complete probability space equipped with a filtration $\mathbb{F} = \{\mathcal{F}_t\}_{t \in [0,T]}$ satisfying the usual conditions. Let $\mathcal{X} = \mathbb{R}^D$ denote the ambient data space and let $\mathcal{Z} \subseteq \mathbb{R}^{d_z}$ denote the latent space, with $d_z \ll D$. We distinguish the full-dimensional reference endpoint $\mathbf{Y}_T$ from the decoder endpoint $\mathbf{X}_T^{\mathrm{dec}} = G_{\boldsymbol{\psi}}(\mathbf{Z})$.

**The Forward Diffusion Bridge.** We define the forward process as a solution to an *stochastic differential equation* (SDE) constrained by boundary conditions. Unlike standard diffusion models where the terminal condition is implicit, the DBAE framework imposes a *pinned* boundary condition $\mathbf{X}_T$. Let the unconditioned reference process $\mathbf{Y}_t$ be an Ornstein–Uhlenbeck (OU) process governed by the *variance-preserving* (VP) SDE:

$$d\mathbf{Y}_t = -\frac{1}{2}\beta(t)\mathbf{Y}_t \, dt + \sqrt{\beta(t)} \, d\mathbf{W}_t, \tag{17}$$

where $\mathbf{W}_t$ is a standard $D$-dimensional Brownian motion adapted to $\mathbb{F}$, and $\beta : [0, T] \to \mathbb{R}^+$ is a continuous noise schedule. The transition density of this unconditioned process is explicitly Gaussian:

$$p_{0t}(\mathbf{y} \mid \mathbf{x}_0) = \phi\big(\mathbf{y}; \sqrt{\bar{\alpha}_t}\mathbf{x}_0, (1 - \bar{\alpha}_t)\mathbf{I}_D\big), \tag{18}$$

where $\phi(\cdot; \boldsymbol{\mu}, \boldsymbol{\Sigma})$ denotes the multivariate Gaussian density and $\bar{\alpha}_t = \exp(-\int_0^t \beta(s)\,\mathrm{d}s)$.

To construct the *diffusion bridge* $\{\mathbf{X}_t\}_{t \in [0,T]}$ targeting a fixed endpoint $\mathbf{x}_T \in \mathcal{X}$, we apply Doob's $h$-transform. The drift term is modified by the logarithmic gradient of the transition kernel:

$$\mathrm{d}\mathbf{X}_t = \left[ -\frac{1}{2}\beta(t)\mathbf{X}_t + \beta(t)\nabla_{\mathbf{X}_t} \log p_{T|t}(\mathbf{x}_T \mid \mathbf{X}_t) \right] \mathrm{d}t + \sqrt{\beta(t)}\,\mathrm{d}\mathbf{W}_t. \tag{19}$$

The term $\nabla_{\mathbf{x}_t} \log p_{T|t}(\mathbf{x}_T \mid \mathbf{x}_t)$ is the *bridge score*. Utilizing the Gaussian property of the OU process, this score admits an analytical closed form that is linear in $\mathbf{X}_t$ and $\mathbf{x}_T$. Consequently, the transition density of the bridge $q_t(\mathbf{x}_t \mid \mathbf{x}_0, \mathbf{x}_T)$ is Gaussian:

$$q_t(\mathbf{x}_t \mid \mathbf{x}_0, \mathbf{x}_T) = \phi(\boldsymbol{\mu}_t(\mathbf{x}_0, \mathbf{x}_T), \boldsymbol{\Sigma}_t), \tag{20}$$

where $\boldsymbol{\mu}_t(\mathbf{x}_0, \mathbf{x}_T)$ is a convex combination of $\mathbf{x}_0$ and $\mathbf{x}_T$, and $\boldsymbol{\Sigma}_t$ depends only on $t$.

**Proof of Capacity Mismatch.** *Step 1: Entropy of the full-dimensional reference endpoint.* For the unconditioned reference endpoint $\mathbf{Y}_T$, the OU process mixes exponentially fast to $\mathcal{N}(\mathbf{0}, \mathbf{I}_D)$. Therefore,

$$\lim_{T \to \infty} H(\mathbf{Y}_T) = \frac{D}{2} \log(2\pi e). \tag{21}$$

This statement concerns the full-dimensional reference endpoint $\mathbf{Y}_T$, not the decoder endpoint $\mathbf{X}_T^{\text{dec}}$.

*Step 2: Entropy of the latent bottleneck.* For the latent variable $\mathbf{Z} \sim \mathcal{N}(\mathbf{0}, \mathbf{I}_{d_z})$, its differential entropy is

$$H(\mathbf{Z}) = \frac{d_z}{2} \log(2\pi e). \tag{22}$$

Thus, the asymptotic stochastic-capacity gap between the full-dimensional reference endpoint and the latent bottleneck is

$$\Delta_{\text{cap}} := H(\mathbf{Y}_T) - H(\mathbf{Z}) \longrightarrow \frac{D - d_z}{2} \log(2\pi e), \qquad d_z \ll D. \tag{23}$$

This expression is dimensionally consistent because both terms are differential entropies measured in nats.

*Step 3: Support mismatch.* If the decoder $G_{\boldsymbol{\psi}} : \mathbb{R}^{d_z} \to \mathbb{R}^D$ is deterministic and sufficiently regular, then the induced decoder endpoint $\mathbf{X}_T^{\text{dec}} = G_{\boldsymbol{\psi}}(\mathbf{Z})$ is supported on an image set of dimension at most $d_z$. When $d_z < D$, this distribution is singular with respect to the $D$-dimensional Lebesgue measure. In contrast, the reference endpoint $\mathbf{Y}_T$ has a full-dimensional Gaussian density. Hence, the overload phenomenon should be interpreted as a structural support-and-capacity mismatch rather than as a finite KL value between two ordinary $D$-dimensional densities:

$$\text{supp}(P_{\text{dec}}) \subseteq G_{\boldsymbol{\psi}}(\mathbb{R}^{d_z}), \qquad \dim G_{\boldsymbol{\psi}}(\mathbb{R}^{d_z}) \leq d_z < D. \tag{24}$$

Therefore, a low-dimensional latent bottleneck cannot faithfully absorb the full stochasticity of the diffusion endpoint without allocating capacity to nuisance variation. This establishes the information overload trap in the high-noise bridge limit. $\square$

### A.4. Proof of Theorem 2 (ASR and Jacobian Regularization)

In this section, we provide the formal derivation connecting the asymmetric semantic rectification (ASR) objective to Jacobian regularization, as stated in Theorem 2. We show that minimizing the reconstruction error under additive Gaussian noise is locally equivalent to imposing a penalty on the Frobenius norm of the encoder's Jacobian matrix.

**Setup and Definitions.** Recall the definition of the student branch input at timestep $t$:

$$\mathbf{x}_t^{\mathcal{S}} = \underbrace{\alpha_t \mathbf{v}^{\mathcal{S}}}_{:=\boldsymbol{\mu}_t} + \sigma_t \boldsymbol{\epsilon}, \quad \text{where } \boldsymbol{\epsilon} \sim \mathcal{N}(\mathbf{0}, \mathbf{I}_D). \tag{25}$$

Here, $\boldsymbol{\mu}_t \in \mathbb{R}^D$ represents the deterministic signal component (the augmented view scaled by $\alpha_t$), and $\sigma_t$ controls the noise magnitude. Let $E_\theta : \mathbb{R}^D \to \mathbb{R}^{d_h}$ denote the student encoder. The target representation $\mathbf{z}^{\mathcal{T}}$ is generated by the teacher branch and is independent of the student's noise $\boldsymbol{\epsilon}$. The ASR objective function is given by:

$$\mathcal{L}_{\text{ASR}} = \mathbb{E}_{\boldsymbol{\epsilon}} \left[ \left\| \mathbf{z}^{\mathcal{T}} - E_\theta(\mathbf{x}_t^{\mathcal{S}}) \right\|_2^2 \right]. \tag{26}$$

**Proof.** We perform a second-order Taylor expansion of the perturbed output $E_\theta(\mathbf{x} + \boldsymbol{\delta})$ around the conditional mean $\mathbf{m}_t = \boldsymbol{\mu}_t$, where $\boldsymbol{\delta} = \sigma_t \boldsymbol{\epsilon}$:

$$E_\theta(\mathbf{m}_t + \boldsymbol{\delta}) = E_\theta(\mathbf{m}_t) + \mathbf{J}_{E_\theta}(\mathbf{m}_t)\boldsymbol{\delta} + \frac{1}{2}\mathbf{H}_{E_\theta}(\mathbf{m}_t)[\boldsymbol{\delta}, \boldsymbol{\delta}] + \mathcal{O}(\|\boldsymbol{\delta}\|_2^3), \tag{27}$$

where $\mathbf{J}_{E_\theta}(\mathbf{m}_t) \in \mathbb{R}^{d_h \times D}$ is the Jacobian matrix and $\mathbf{H}_{E_\theta}(\mathbf{m}_t)$ denotes the collection of component-wise Hessians $\{\nabla^2 E_{\theta,j}(\mathbf{m}_t)\}_{j=1}^{d_h}$, viewed as a bilinear map. We analyze the squared Euclidean norm of the error $\mathbf{r} = \mathbf{z}^{\mathcal{T}} - E_\theta(\mathbf{m}_t)$:

$$\mathcal{L}_{\text{ASR}} = \mathbb{E}_{\boldsymbol{\epsilon}} \left[ \left\| \mathbf{r} - \mathbf{J}_{E_\theta}(\mathbf{m}_t)\boldsymbol{\delta} \right\|_2^2 \right]$$
$$+ \mathcal{O}\left( \sigma_t^2 \|\mathbf{r}\|_2 \|\mathbf{H}_{E_\theta}(\mathbf{m}_t)\|_{\text{op}} + \sigma_t^4 \right). \tag{28}$$

Expanding the norm:

$$\mathcal{L}_{\text{ASR}} = \mathbb{E}_{\boldsymbol{\epsilon}} \left[ \mathbf{r}^\top \mathbf{r} - 2\mathbf{r}^\top \mathbf{J}_{E_\theta}(\mathbf{m}_t)\boldsymbol{\delta} + \boldsymbol{\delta}^\top \mathbf{J}_{E_\theta}(\mathbf{m}_t)^\top \mathbf{J}_{E_\theta}(\mathbf{m}_t)\boldsymbol{\delta} \right]. \tag{29}$$

Using the linearity of expectation and properties of $\boldsymbol{\delta} = \sigma_t \boldsymbol{\epsilon}$ with $\boldsymbol{\epsilon} \sim \mathcal{N}(\mathbf{0}, \mathbf{I}_D)$: 1. $\mathbb{E}[\mathbf{r}^\top \mathbf{r}] = \|\mathbf{z}^{\mathcal{T}} - E_\theta(\mathbf{m}_t)\|_2^2$ (Alignment term). 2. $\mathbb{E}[-2\mathbf{r}^\top \mathbf{J}_{E_\theta}(\mathbf{m}_t)\boldsymbol{\delta}] = -2\mathbf{r}^\top \mathbf{J}_{E_\theta}(\mathbf{m}_t)\sigma_t \mathbb{E}[\boldsymbol{\epsilon}] = 0$. 3. For the quadratic term, using the trace trick $\mathbf{a}^\top \mathbf{M} \mathbf{a} = \text{Tr}(\mathbf{M}\mathbf{a}\mathbf{a}^\top)$:

$$\mathbb{E}\left[ \boldsymbol{\delta}^\top \mathbf{J}_{E_\theta}(\mathbf{m}_t)^\top \mathbf{J}_{E_\theta}(\mathbf{m}_t)\boldsymbol{\delta} \right] = \mathbb{E}\left[ \text{Tr}\left( \mathbf{J}_{E_\theta}(\mathbf{m}_t)^\top \mathbf{J}_{E_\theta}(\mathbf{m}_t)\boldsymbol{\delta}\boldsymbol{\delta}^\top \right) \right] \tag{30}$$

$$= \text{Tr}\left( \mathbf{J}_{E_\theta}(\mathbf{m}_t)^\top \mathbf{J}_{E_\theta}(\mathbf{m}_t)\mathbb{E}[\boldsymbol{\delta}\boldsymbol{\delta}^\top] \right) \tag{31}$$

$$= \text{Tr}\left( \mathbf{J}_{E_\theta}(\mathbf{m}_t)^\top \mathbf{J}_{E_\theta}(\mathbf{m}_t)(\sigma_t^2 \mathbf{I}_D) \right) \tag{32}$$

$$= \sigma_t^2 \text{Tr}\left( \mathbf{J}_{E_\theta}(\mathbf{m}_t)^\top \mathbf{J}_{E_\theta}(\mathbf{m}_t) \right) = \sigma_t^2 \|\mathbf{J}_{E_\theta}(\mathbf{m}_t)\|_F^2. \tag{33}$$

Combining these terms, we obtain:

$$\mathcal{L}_{\text{ASR}} = \|\mathbf{z}^{\mathcal{T}} - E_\theta(\mathbf{m}_t)\|_2^2 + \sigma_t^2 \|\mathbf{J}_{E_\theta}(\mathbf{m}_t)\|_F^2 + \mathcal{O}\left( \sigma_t^2 \|\mathbf{r}\|_2 \|\mathbf{H}_{E_\theta}(\mathbf{m}_t)\|_{\text{op}} + \sigma_t^4 \right). \tag{34}$$

This confirms that minimizing the ASR loss under perturbation effectively imposes a penalty on the Frobenius norm of the encoder's Jacobian, smoothing the learned manifold. When the aligned residual $\|\mathbf{r}\|_2$ is small, the leading non-constant term is precisely the dynamic Jacobian penalty. $\qquad \square$

### A.5. Contrastive Rectification versus Normalized MSE

A natural alternative to ArcDAE's contrastive rectification is normalized mean squared error (Norm-MSE) between teacher and student features. However, this objective is insufficient under extreme diffusion noise.

The ASR analysis shows that minimizing the alignment error around the conditional mean $\mathbf{m}_t$ yields the residual decomposition

$$\mathcal{L}_{\text{align}} = \|E_\theta(\mathbf{m}_t) - \mathbf{z}^{\mathcal{T}}\|_2^2 + \sigma_t^2 \|\mathbf{J}_{E_\theta}(\mathbf{m}_t)\|_F^2 + \mathcal{R}_{\text{align}}(t),$$

where, for a local constant $C > 0$ independent of sufficiently small $\sigma_t$,

$$|\mathcal{R}_{\text{align}}(t)| \leq C\left( \sigma_t^2 \|E_\theta(\mathbf{m}_t) - \mathbf{z}^{\mathcal{T}}\|_2 \|\mathbf{H}_{E_\theta}(\mathbf{m}_t)\|_{\text{op}} + \sigma_t^4 \right).$$

This Jacobian regularization is beneficial because it suppresses high-frequency nuisance variation. Yet when $t \to T$, the corruption level is severe and the coefficient $\sigma_t^2$ can dominate the optimization. If alignment is implemented only with unconstrained Norm-MSE, the model can minimize the objective by driving

$$\|\mathbf{J}_{E_\theta}(\mathbf{m}_t)\|_F^2 \to 0,$$

which makes the encoder locally insensitive to input variations. On the hypersphere, this corresponds to a locally constant mapping and may lead to dimensional collapse.

InfoNCE avoids this failure mode by combining alignment and uniformity. The numerator preserves semantic consistency between the corrupted student and the clean teacher anchor, while the denominator repels different samples and keeps the feature covariance well-conditioned. This realizes the useful local smoothing predicted by ASR without allowing the representation to collapse globally. This perspective is consistent with the alignment-uniformity interpretation of contrastive learning (Wang & Isola, 2020).

*Table 7.* Ablation on the ASR objective function at 330K training steps. Bold indicates the best result, and underline with very light cyan indicates the second-best result.

| Dataset | Metric | Normalized MSE | ArcDAE (InfoNCE) |
|---------|--------|---------------|------------------|
| FFHQ | AP ($\uparrow$) | 0.532 | **0.585** |
| | $r$ ($\uparrow$) | 0.537 | **0.598** |
| | MSE ($\downarrow$) | 0.435 | **0.412** |
| CelebA | AP ($\uparrow$) | 0.524 | **0.573** |
| | $r$ ($\uparrow$) | 0.515 | **0.550** |
| | MSE ($\downarrow$) | 0.439 | **0.420** |

# B. Implementation Details

In this section, we provide rigorous specifications of the ArcDAE architecture components, the mathematical formulation of the TAPA augmentation pipeline, and the hyperparameters used in our experiments.

### B.1. Model Architecture

Our framework consists of three coupled networks: a Semantic Encoder, a Semantic Generator (Decoder), and a Conditional Denoising U-Net. The backbone design adheres to the configurations in (Dhariwal & Nichol, 2021) and (Kim et al., 2025).

**1. Semantic Encoder** ($E_\phi$)**:** We employ the *BeatGANs* encoder architecture, optimized for capturing high-fidelity semantic representations.

- **Structure:** The network processes inputs $\mathbf{x} \in \mathbb{R}^{3 \times 256 \times 256}$ through a sequence of Residual Blocks with channel multipliers $[1, 1, 2, 2, 4, 4]$. Each block utilizes Group Normalization ($G = 32$) and SiLU activation.

- **Downsampling:** Unlike standard strided convolutions, we utilize *Average Pooling* for downsampling to preserve spatial semantics more effectively before the bottleneck.

- **Latent Bottleneck:** The final spatial feature map $\mathbf{h}_{enc} \in \mathbb{R}^{512 \times 8 \times 8}$ undergoes a specific transformation sequence: GroupNorm $\to$ SiLU $\to$ Conv$_{3\times3}$ $\to$ Flatten $\to$ Linear Projection. This yields the compact semantic latent $\mathbf{z} \in \mathbb{R}^{512}$.

**2. Semantic Generator / Decoder** ($G_\psi$)**:** This module is responsible for predicting the terminal state $\mathbf{x}_T$ from the latent $\mathbf{z}$. It mirrors the encoder structure but operates in reverse.

- **Projection:** The latent $\mathbf{z}$ is first projected and reshaped to $\mathbb{R}^{512 \times 4 \times 4}$.

- **Upsampling:** We employ nearest-neighbor interpolation followed by convolution for upsampling. The channel multipliers follow the sequence $[4, 4, 2, 2, 1, 1]$.

- **Output:** The final layer maps features to $\mathbb{R}^{3 \times 256 \times 256}$, producing the semantic guess $\mathbf{x}_T$.

**3. Denoising U-Net** ($\epsilon_\theta$)**:** The core diffusion backbone is an ADM U-Net conditioned on $\mathbf{z}$.

- **Conditioning (AdaGN):** The semantic latent $\mathbf{z}$ is injected into every residual block via Adaptive Group Normalization (AdaGN):

$$\text{AdaGN}(\mathbf{h}, \mathbf{z}) = \mathbf{y}_s(\mathbf{z}) \cdot \text{GroupNorm}(\mathbf{h}) + \mathbf{y}_b(\mathbf{z}), \tag{35}$$

where $\mathbf{y}_s$ and $\mathbf{y}_b$ are linear projections of $\mathbf{z}$ regulating scale and shift.

**Dual Projectors for Timestep-adaptive Granularity Alignment (TGA):** To explicitly decouple semantic abstraction from spatial granularity, we attach two distinct projection heads to the Denoising U-Net feature hierarchy. The detailed specifications are provided in Tables 8 and 9.

- **Global (Holistic) Projector** ($\mathcal{P}_{\text{hol}}$)**:** Designed to enforce strict translation invariance. It processes features from the *Middle Block* and employs a Global Average Pooling (GAP) bottleneck sandwiched between normalization and projection layers (Table 8) to strictly compress spatial information into a global semantic vector.

- **Local (Partial) Projector** ($\mathcal{P}_{\text{par}}$)**:** Designed to retain spatial sensitivity. It branches from the penultimate encoder layer (Layer $[-2]$). We implement an *Attention Pooling* mechanism (Radford et al., 2021) (Table 9) where a learnable query dynamically aggregates spatial features, allowing the model to focus on saliency-induced density regions.

*Table 8.* Specification of the Global (Holistic) Projector. This module transforms the Middle Block features into a compact, translation-invariant semantic vector.

| LAYER / OPERATION | INPUT DIM | OUTPUT DIM | CONFIGURATION |
|---|---|---|---|
| INPUT (MIDDLE BLOCK) | $512 \times 8 \times 8$ | – | – |
| GROUP NORMALIZATION | $512$ | $512$ | GROUPS=32 |
| ACTIVATION | $512$ | $512$ | SILU |
| ADAPTIVE AVGPOOL | $512 \times 8 \times 8$ | $512 \times 1 \times 1$ | SPATIAL COMPRESSION |
| LINEAR PROJECTION | $512$ | $512$ | CONV $1 \times 1$ |
| FLATTEN | $512 \times 1 \times 1$ | $512$ | – |
| LAYER NORMALIZATION | $512$ | $512$ | FINAL ALIGNMENT |

*Table 9.* Specification of the Local (Partial) Projector. Utilizing Attention Pooling to capture fine-grained spatial dependencies from Encoder Layer $[-2]$.

| LAYER / OPERATION | INPUT DIM | OUTPUT DIM | CONFIGURATION |
|---|---|---|---|
| INPUT (LAYER -2) | $C_{enc} \times 16 \times 16$ | – | – |
| GLOBAL TOKEN INIT | $C_{enc} \times 16 \times 16$ | $C_{enc}$ | MEAN POOLING |
| POSITIONAL EMBED | $(16^2 + 1) \times C_{enc}$ | $(16^2 + 1) \times C_{enc}$ | LEARNABLE |
| QKV PROJECTION | $C_{enc}$ | $3 \times C_{enc}$ | LINEAR |
| MULTI-HEAD ATTENTION | $C_{enc}$ | $C_{enc}$ | HEADS=64 |
| OUTPUT PROJECTION | $C_{enc}$ | $512$ | LINEAR |
| GLOBAL TOKEN EXTRACT | $(16^2 + 1) \times 512$ | $512$ | INDEX $[0]$ |
| LAYER NORMALIZATION | $512$ | $512$ | FINAL ALIGNMENT |

### B.2. Timestep-Aware Progressive Augmentation (TAPA)

TAPA acts as a dynamic augmentation mechanism that prevents the encoder from learning shortcut solutions while staying consistent with the main-text definition and the ablation tables. We define the augmentation intensity as a linear function of the diffusion timestep, $\gamma(t) = t/T$, where $T = 1000$. The student augmentation operator is

$$\Psi_t = \psi_{\text{jitter},t} \circ \psi_{\text{blur},t} \circ \psi_{\text{crop},t}, \tag{36}$$

which means that the input is sequentially processed by cropping, Gaussian blurring, and color jittering. The augmented view and the subsequent diffusion-corrupted student input are therefore

$$\begin{aligned} \mathbf{x}_{\text{aug}} &= \Psi_t(\mathbf{x}_0), \\ \mathbf{x}_t^{\mathcal{S}} &= \sqrt{\bar{\alpha}_t} \, \mathbf{x}_{\text{aug}} + \sqrt{1 - \bar{\alpha}_t} \, \boldsymbol{\epsilon}, \qquad \boldsymbol{\epsilon} \sim \mathcal{N}(\mathbf{0}, \mathbf{I}_D). \end{aligned} \tag{37}$$

Thus, additive Gaussian noise belongs to the diffusion corruption step rather than to the TAPA operator itself.

The specific parameter spaces for each transformation are:

1. **Progressive Resized Crop ($\psi_{\text{crop},t}$):**

   - Crop scale $s \in [s_{\min}(t), 1.0]$.
   - $s_{\min}(t) = 1.0 - 0.8 \cdot \gamma(t)$. As $t \to T$, the student receives only approximately 20% of the original spatial evidence, distributed across sparse blocks in our implementation, necessitating semantic completion rather than literal recovery from a single crop.
   - **Distributed Implementation:** We clarify that the augmentation intensity scales linearly with the randomly sampled timestep $t$. Severe cropping occurs only at large diffusion timesteps ($t \to T$). To prevent complete structural loss, our implementation utilizes a multi-block distributed crop rather than a single monolithic bounding box, which functionally preserves sparse global layout hints even at these extreme scales.

2. **Gaussian Blur ($\psi_{\text{blur},t}$):**

   - Kernel size is fixed at $23 \times 23$.
   - The blur standard deviation is $\sigma_{\text{blur}}(t) \in [0.1, 0.1 + 2.0 \cdot \gamma(t)]$. The blur intensity increases with $t$, removing high-frequency textures that typically leak identity information.

3. **Color Jitter ($\psi_{\text{jitter},t}$):**

   - Brightness, contrast, and saturation factors satisfy $\kappa_i \in [1 - \eta_{\text{jitter}}\gamma(t), 1 + \eta_{\text{jitter}}\gamma(t)]$ for $i \in \{1, 2, 3\}$.
   - We use $\eta_{\text{jitter}} = 0.4$ by default, which discourages the student from exploiting low-level color statistics.

Note that the teacher encoder always receives a *weak* view, implemented as a horizontal flip, to provide a stable semantic anchor $\mathbf{z}_{\text{anchor}}$.

**Operator Composition and Order.** The sequence $\Psi_t = \psi_{\text{jitter},t} \circ \psi_{\text{blur},t} \circ \psi_{\text{crop},t}$ is intentionally designed. Applying Crop before Blur ensures the blur kernel operates only on the valid zoomed-in patch, which helps avoid artificial boundary artifacts. Additionally, for the Progressive Resized Crop, our implementation utilizes a multi-block distributed crop rather than a single monolithic bounding box. This functionally preserves sparse global layout hints even at extreme scales where up to 80% of the spatial area is discarded.

### B.3. Training Hyperparameters

We summarize the default hyperparameters used for FFHQ and CelebA experiments in Table 10.

*Table 10.* Hyperparameters for ArcDAE training.

| Hyperparameter | Value |
|---|---|
| Diffusion Steps ($T$) | 1000 |
| Noise Schedule | Linear ($\beta_{\text{start}} = 10^{-4}, \beta_{\text{end}} = 0.02$) |
| Optimizer | AdamW |
| Learning Rate | $1 \times 10^{-4}$ |
| Batch Size | 128 |
| Latent Dimension ($d_z$) | 512 |
| Student EMA ($m$) | 0.9999 |
| Teacher EMA ($m$) | 0.999 |
| Contrastive Temp ($\tau_{\text{nce}}$) | 0.07 |
| Balance Weight ($\lambda_{\text{align}}$) | 0.1 for FFHQ; 0.01 for CelebA |
| Warm-up Steps ($N_{\text{warm}}$) | 20,000 |
| Total Training Steps | 1,020,000 |

### B.4. Architectural Design Rationale

**Maintaining the Diffusion Decoder.** The diffusion objective provides a necessary *spatial inductive bias*. Unlike pure SSL (e.g., SimCLR) which allows the latent to lose spatial coordinates, the reconstruction requirement forces $\mathbf{z}$ to retain layout information (e.g., eye position), which is critical for dense prediction tasks.

**Stochastic DBAE versus Deterministic DBAE.** The stochastic variant of DBAE introduces randomness in the encoder $q(\mathbf{z}|\mathbf{x})$. This acts as implicit regularization (passive noise filtering). ArcDAE replaces this *blind* filtering with *active* rectification (TAPA + Teacher), yielding the performance gains observed in Table 1.

## C. Additional Generalization and Robustness Experiments

### C.1. Generalization Beyond Face-Centric Benchmarks

To verify that ArcDAE is not restricted to face-centric distributions, we evaluate it on CIFAR-10 for semantic separability and LSUN Horse/Bedroom for generative reconstruction. These results demonstrate that ArcDAE improves semantic AUROC and achieves stronger SSIM on diverse object categories and complex backgrounds. The MSE values remain comparable but are not the best, which is consistent with the perception-distortion trade-off discussed in Sec. 4.3.

*Table 11.* Generalization on non-face datasets. Latent dimension is 512. Bold indicates the best result, and underline indicates the second-best result.

| Method | CIFAR-10 AUROC ↑ | LSUN Horse SSIM ↑ / MSE ↓ | LSUN Bedroom SSIM ↑ / MSE ↓ |
|---|---|---|---|
| DiffuseVAE / DiffAE | 0.736 | 0.857 / 0.025 | 0.910 / 0.017 |
| DBAE | 0.836 | 0.902 / **0.012** | 0.948 / **0.007** |
| ArcDAE | **0.861** | **0.918** / 0.014 | **0.956** / 0.008 |

### C.2. Feature Stability under Controlled Perturbations

We inject high-frequency Gaussian perturbations $\boldsymbol{\epsilon} \sim \mathcal{N}(\mathbf{0}, \sigma_{\mathrm{p}}^2 \mathbf{I}_D)$ into FFHQ validation inputs and measure latent cosine similarity between clean and perturbed global semantic representations $\mathbf{z}_{\mathrm{holo}}$. ArcDAE maintains substantially higher latent stability, especially under severe corruption.

*Table 12.* Latent cosine similarity under controlled high-frequency Gaussian perturbations on FFHQ. Bold indicates the best result, and underline with very light cyan indicates the second-best result.

| Perturbation $\sigma_{\mathrm{p}}$ | ArcDAE | DBAE | PDAE | DiffAE |
|---|---|---|---|---|
| 0.05 | **0.975** | 0.882 | 0.854 | 0.812 |
| 0.10 | **0.932** | 0.745 | 0.701 | 0.645 |
| 0.20 | **0.841** | 0.458 | 0.412 | 0.318 |

### C.3. Mechanism of Catastrophic Failure in TAPA (No Teacher)

To understand the catastrophic failure observed in the *TAPA (No Teacher)* ablation, we note that the model does not simply ignore the input but suffers from dimensional collapse.

Under the standard TAPA protocol, timesteps are randomly sampled. When the model encounters large diffusion timesteps $(t \rightarrow T)$, the progressive crop operation discards up to 80% of the spatial area. Without the frozen teacher acting as a stable semantic anchor to rectify these extreme samples, the information available from the corrupted input is substantially reduced rather than literally zero. Repeated high-noise samples therefore bias the unconstrained student encoder toward a low-rank, locally contractive representation, as reflected by the reduced latent variance and SVD entropy in Table 13.

*Table 13.* Dimensional collapse analysis at 500k steps on FFHQ. Bold indicates the best result, underline with very light cyan indicates the second-best result, and light-red cells mark the collapsed variant.

| Metric | ArcDAE (Full) | TAPA (No Teacher) | DBAE-d |
|---|---|---|---|
| AP (Linear Probe) | **0.658** | 0.205 | 0.623 |
| Latent Variance | **0.85** | 0.26 | 0.81 |
| SVD Entropy | **4.45** | 2.82 | 4.25 |

## C.4. TAPA Composition and Hyperparameter Sensitivity

Progressive Crop and Gaussian Blur are the dominant operators in TAPA. The order is designed as

$$\Psi_t = \psi_{\text{jitter},t} \circ \psi_{\text{blur},t} \circ \psi_{\text{crop},t},$$

so that Crop is applied before Blur, ensuring that the blur kernel operates on the valid zoomed-in patch and avoids artificial boundary artifacts.

*Table 14.* Operator composition and order ablation on FFHQ. Bold indicates the best result, and underline with very light cyan indicates the second-best result.

| TAPA Configuration | AP ↑ | LPIPS ↓ | FID ↓ |
|---|---|---|---|
| Full TAPA (Crop → Blur → Jitter) | **0.681** | **0.065** | **10.28** |
| Crop → Blur (No Jitter) | 0.675 | 0.067 | 10.46 |
| Reverse Order (Jitter → Blur → Crop) | 0.671 | 0.068 | 10.65 |

*Table 15.* Hyperparameter sensitivity for TAPA and SNR-based weighting on FFHQ. Bold indicates the best result, and underline with very light cyan indicates the second-best result.

| Ablation Target | Variant 1 | Default | Variant 2 |
|---|---|---|---|
| TAPA Min Scale $s_{\min}$ | 0.08 → 0.675 | 0.20 → **0.681** | 0.50 → 0.672 |
| TAPA Max Blur $\Delta\sigma_{\text{blur}}$ | 1.00 → 0.672 | 2.00 → **0.681** | 3.00 → 0.677 |
| SNR Temp $\tau_{\text{snr}}$ | 0.50 → 0.678 | 1.00 → **0.681** | 2.00 → 0.676 |

## C.5. Computational Overhead

ArcDAE introduces minimal overhead relative to DBAE-style baselines. The teacher branch is stop-gradient and does not require activation storage for backpropagation. Using the same U-Net backbone and batch size 128, the additional trainable parameters come only from lightweight projection heads for contrastive rectification.

*Table 16.* Training cost comparison under the same backbone and batch size.

| Method | Trainable Params | Train Time (ms/img) |
|---|---|---|
| DDIM | 99M | 9.687 |
| DiffAE | 129M | 12.088 |
| PDAE | 280M | 12.163 |
| DBAE | 161M | 13.190 |
| ArcDAE | **163M** | **13.590** |

Compared with DBAE, ArcDAE increases trainable parameters by only about 1.2% and training time by about 3%. The peak VRAM increase is also +1.9GB over DBAE's 25.7GB, corresponding to a relative increase of approximately 7.4%, making the improved semantic separability and convergence efficiency a favorable trade-off.

## C.6. Impact of Latent Space Quality on LDM Prior Training

The quality of the learned latent space directly affects the efficiency of the latent diffusion prior $p_\omega(\mathbf{z})$. By filtering out high-entropy stochastic noise, ArcDAE produces a smoother and more linearly separable latent manifold. In contrast, unconstrained bridges such as DBAE entangle high-frequency nuisance variation, making the latent distribution more irregular and harder to model.

*Table 17.* LDM prior training efficiency under restricted capacity. Bold indicates the best result.

| Latent Source | Prior Architecture | Epochs | FID ↓ |
|---|---|---|---|
| DBAE-d | 80M Light U-Net | ≥ 250 (struggling) | 13.50 |
| ArcDAE | 80M Light U-Net | **90** | **10.45** |

