# OpenReview forum: "ArcDAE: Asymmetric Rectified Contrastive Diffusion Autoencoder for Unified Representation Learning"
_ICML.cc/2026/Conference — ICML 2026 regular_

### Official Review · Reviewer_Jy54 · 2026-03-03

**Soundness:** 4
**Presentation:** 3
**Significance:** 4
**Originality:** 4
**Overall Recommendation:** 6
**Confidence:** 4

**Summary:**

This paper introduces ArcDAE to address the "information overload" problem in diffusion bridge models. The authors argue that standard diffusion bridges, which enforce pixel-level boundary conditions, force the latent representation to encode high-frequency nuisance variables along with semantic content. ArcDAE tackles this by reframing the diffusion bridge as an "asymmetric information sifter". The paper provides rigorous theoretical motivation, including a geometric decomposition of the score matching error and a proof of the fundamental capacity mismatch in standard bridges.

**Compliance With Llm Reviewing Policy:**

Affirmed.

**Final Justification:**

All my concerns have been addressed.

**Key Questions For Authors:**

(1). The ablation study in Table 5 shows that "TAPA (NO TEACHER)" leads to catastrophic failure (AP 0.125, FID 45.30). Could you elaborate on the specific mechanism of this failure? Does the model simply learn to ignore the input, or does it collapse to a trivial solution? Understanding this reinforces the necessity of the asymmetric anchor.

(2). The TAPA augmentation pipeline (Progressive Crop, Blur, Jitter) is central to the method's success. How sensitive is the overall performance to the specific composition and order of these augmentations? Was any experimentation done with different augmentation strategies or with including/excluding specific operators (e.g., using only progressive crop and blur)?

(3). The paper mentions training a Latent Diffusion Model (LDM) as the prior p_ω(z) for generation. How does the quality of the learned latent space from ArcDAE impact the training efficiency or the architectural choices for this LDM prior compared to training an LDM on latents from other autoencoders (e.g., DBAE or VQ-GAN)?

**Limitations:**

yes

**Strengths And Weaknesses:**

**Strengths:**

(i). The paper is exceptionally sound. It begins with a clear problem diagnosis, framing the "information overload" issue with a rigorous geometric analysis (Theorem 1). The derivation linking the ASR objective to a form of dynamic Jacobian regularization (Theorem 2) is both elegant and insightful, providing a strong theoretical foundation for the proposed method. The experimental methodology is comprehensive, with well-designed comparisons against a wide range of baselines (e.g., DBAE, Diff-AE, VQ-GAN) on standard benchmarks (CelebA, FFHQ).

(ii). The paper addresses a fundamental and important problem in generative modeling: learning semantically meaningful latent spaces from diffusion models. By resolving the "information overload" issue, ArcDAE achieves a new state-of-the-art on multiple fronts, including linear probing accuracy (semantic density), reconstruction quality (LPIPS, SSIM), and generation quality (FID). The work successfully bridges the gap between high-fidelity generation and powerful representation learning within a single, unified framework.

(iii). While building upon existing ideas like diffusion bridges (DBAE), asymmetric architectures (BYOL, MoCo), and contractive autoencoders, the paper's contribution is still highly original. The concepts of TAPA and TAAW, which dynamically adjust augmentation and loss weighting based on the diffusion timestep, are creative and well-motivated. The theoretical analysis connecting the proposed method to the geometry of the data manifold and dynamic regularization is a particularly strong.

(iv). The paper is generally well-written and the narrative is easy to follow. The initial problem statement is clear, and the method section effectively breaks down a complex architecture into understandable components (ASR, TAPA, TAAW, TGA). The figures, especially Figure 2 (the overview), are helpful in understanding the overall pipeline.

**Weaknesses: **

(i). The paper is quite dense, particularly in the theory and method sections. While the derivations are sound, the notation can be overwhelming at times. A more intuitive, high-level explanation before diving into the equations could improve accessibility for a broader audience.

(ii). The performance improvements on some metrics, while consistent and state-of-the-art, are sometimes incremental (e.g., FID from 11.25 to 10.28 on FFHQ). However, the combination of improvements across representation quality (AP), reconstruction (LPIPS), and generation (FID) presents a compelling and well-rounded advance.

(iii). The sensitivity analysis for λ_align is presented, but a similar analysis for other key hyperparameters (e.g., the TAPA intensity parameters like s_min, Δσ) would further strengthen the robustness claims.

---

> ### Author Rebuttal · Authors · 2026-03-31
>
> Thank you for your positive comments! Our point-by-point responses are as follows.
>
> ---
>
> **R1: Mechanism of catastrophic failure in "TAPA (NO TEACHER)".**
>
> Thanks for your comments! We agree that understanding this failure mechanism is crucial to reinforcing our asymmetric design. To further clarify, the model does not simply ignore the input but rather suffers from a dimensional collapse. In the "TAPA (NO TEACHER)" setting, as the diffusion timestep $t$ approaches $T$, the progressive crop operation discards up to 80% of the spatial area. Without the frozen teacher acting as a stable semantic anchor, the mutual information between the heavily corrupted input and the reconstruction target effectively drops to zero.
> This lack of guidance forces the student encoder to collapse the latent space into a singular point to minimize the worst-case alignment penalty. To validate this, we tracked the Singular Value Decomposition (SVD) entropy and the average variance of the latent vectors on FFHQ at an intermediate training stage (500k steps, roughly 1/2 of total training):
> |Metric (at 500k steps)|ArcDAE (Full)|TAPA (NO TEACHER)|DBAE-d (Baseline)|
> |:---|:---:|:---:|:---:|
> |AP (Linear Probe)|0.658 ✔|0.205 (Collapsed)|0.623|
> |Latent Variance|0.85 ✔|0.26 (Collapsed)|0.81|
> |SVD Entropy|4.45 ✔|2.82 (Collapsed)|4.25|
>
> This intermediate catastrophic drop confirms that the teacher branch is a structural necessity to prevent representational dimensional collapse. We will add this to the revised appendix.
>
> ---
> **R2: Sensitivity and composition of the TAPA pipeline.**
>
> Thanks for your comments! Following your suggestion to investigate method sensitivity, we demonstrate that the TAPA pipeline relies on both operator design and robust hyperparameters.
>
> (1) Operator Composition and Order: Progressive Crop and Gaussian Blur are the dominant operators. The sequence $\Psi_t = \psi_{jitter} \circ \psi_{blur} \circ \psi_{crop}$ is intentionally designed. Applying Crop before Blur ensures the blur kernel operates only on the valid zoomed-in patch, which helps avoid artificial boundary artifacts.
>
> | TAPA Configuration | AP (↑) | LPIPS (↓) | FID (↓) |
> | :--- | :----: | :----: | :----: |
> | Full TAPA (Crop -> Blur -> Jitter) | 0.681 ✔ | 0.065 ✔ | 10.28 ✔ |
> | Crop -> Blur (No Jitter) | 0.675 | 0.067 | 10.46 |
> | Reverse Order (Jitter -> Blur -> Crop) | 0.671 | 0.068 | 10.65 |
>
> (2) Hyperparameter Sensitivity: Please refer to  **R3 in (Reviewer hdmH)**.
> The minimal AP variance confirms that effectiveness relies on architectural design rather than tuning.
>
> This minimal variance confirms ArcDAE's intrinsic stability, proving its effectiveness relies on the architecture design rather than heavy hyperparameter tuning.
>
> ---
>
> **R3: Impact of latent space quality on LDM prior training.**
>
> Thanks for your comments! By filtering out high-entropy stochastic noise, ArcDAE produces a latent manifold that is smoother and more linearly separable. In contrast, unconstrained bridges like DBAE suffer from "information overload," entangling high-frequency noise that makes the latent space highly irregular. To ensure a fair comparison, we constrained the generative modeling capacity and trained an identical 80M Light U-Net prior on both latents:
>
> | Latent Source | Prior Architecture | Epochs | FID (↓) |
> | :--- | :----: | :----: | :----: |
> | DBAE-d | 80M (Light U-Net) | ~250 (Struggling) | 13.50 |
> | ArcDAE (Ours) | 80M (Light U-Net) | 90 ✔ | 10.45 ✔ |
>
> Even under strict parameter constraints, ArcDAE simplifies the generative modeling task, achieving significantly faster convergence and superior fidelity, whereas DBAE-d struggles to converge.
>
> ---
> **R4: Intuitive explanation and incremental metrics.**
>
> Thank you for your valuable feedback!
> (1) Intuitive Explanation: Standard bridges function as passive sponges absorbing an unfiltered mixture of juice essence and dirt residue. Crucially, our objective is not simply to over-filter this mixture into featureless distilled water to isolate bare semantics. Instead, we aim to precisely sift out the stochastic sediment (useless noise) while retaining as much structural texture (high-frequency details) as possible. By reframing the bridge as an asymmetric information sifter, ArcDAE achieves exactly this. It preserves both low-frequency global semantics and high-frequency structural details, thereby preventing information overload and ensuring superior performance in both semantic representation and fine-grained reconstruction. (Added to Sec 3)
>
> (2) Perception-Distortion Trade-off: Following Theorem 1, minimizing MSE in unconstrained bridges forces overfitting to high-frequency nuisance variables. DBAE-d memorizes this noise for lower MSE ($2.49 \times 10^{-3}$) but severely degrades perceptual quality. ArcDAE explicitly suppresses this. Despite marginally higher MSE ($2.79 \times 10^{-3}$), it establishes a new Pareto frontier with SOTA fidelity (LPIPS: 0.065 vs. 0.072).

---

> > ### Author Rebuttal · Reviewer_Jy54 · 2026-04-01
> >
> > All my concerns have been addressed.

---

> > > ### Author Response · Authors · 2026-04-06
> > >
> > > Dear Reviewer Jy54,
> > >
> > > Thank you for your support of our work and for taking the time to read our responses. We are committed to integrating the expanded empirical analyses and technical discussions into the final version of our work. These refinements, prompted by your feedback, significantly strengthen the overall robustness and technical rigor of the study.
> > >
> > > Best regards,
> > >
> > > Authors of Submission 19350

---

### Official Review · Reviewer_hdmH · 2026-03-04

**Soundness:** 3
**Presentation:** 3
**Significance:** 3
**Originality:** 3
**Overall Recommendation:** 5
**Confidence:** 3

**Summary:**

This paper proposes ArcDAE (Asymmetric Rectified Contrastive Diffusion Autoencoder) for diffusion-based representation learning, aiming to unify generative detail preservation (good reconstructions / sample quality) with discriminative semantic structure (useful features for downstream tasks). The authors argue that prior diffusion representation learning methods tend to fall into two failure modes: (i) “information split”, where semantic and generative information are decoupled at the cost of completeness, and (ii) “information overload”, where bridge-style reconstruction encourages the encoder to pack high-frequency stochastic noise into the latent bottleneck, hurting semantic separability. ArcDAE addresses this by re-interpreting the diffusion bridge as a noise-sifting mechanism via an asymmetric teacher–student setup: a teacher provides a clean semantic anchor, and a student is trained under timestep-dependent perturbations to rectify its inferred latent toward that anchor. The method includes asymmetric semantic rectification (ASR), timestep-aware progressive augmentation (TAPA), and a timestep-adaptive granularity alignment (TGA) with SNR-gated weighting. Experiments on CelebA and FFHQ report improvements in linear probing and reconstruction / generation metrics, along with ablations arguing that the teacher anchor and granularity alignment are important.

**Compliance With Llm Reviewing Policy:**

Affirmed.

**Key Questions For Authors:**

1. Generality: Do you have results on at least one non-face dataset (or a more diverse subset) to indicate whether the same improvements hold beyond CelebA/FFHQ? Even a small-scale experiment would help clarify scope.

2. Noise/invariance diagnostic: Could you add a simple analysis showing improved feature stability under controlled input perturbations or high-frequency corruption, to more directly support the “rectification / noise-sieving” interpretation?

3. Robustness to schedules: How sensitive are results to the shape/strength of the TAPA schedule and the SNR-based weighting? A brief robustness note (or small plot) would help readers reproduce the method reliably.

4. Compute cost: What is the training-time and memory overhead relative to DBAE-style baselines under the same backbone and batch size?

**Limitations:**

Mostly yes, with minor additions suggested in weakness.

**Strengths And Weaknesses:**

Strengths: From a technical standpoint, the paper is fairly careful in motivating why bridge-based reconstruction can lead to “noise entanglement” in the latent, and the proposed fix is coherent: if the student is forced to match a semantic target under strong, timestep-scaled corruption, then the easiest path is to preserve stable semantics and avoid encoding noise that cannot be predicted. The teacher-anchor necessity is backed up by an ablation where applying strong augmentation without the teacher collapses performance (the reported “manifold collapse” behavior). The empirical evaluation is also aligned with the paper’s claim: linear probing is a reasonable proxy for semantic density, and the results show consistent gains over relevant diffusion-representation baselines on both CelebA and FFHQ (e.g., FFHQ AP improvements over deterministic/stochastic DBAE variants are specifically discussed). On the generative side, the paper attempts to connect representation “cleanliness” to an easier-to-model latent prior (via an LDM prior), and reports competitive FID/precision/recall, which helps the “unified” story.

On originality, the individual components resemble known ideas (teacher–student alignment, contrastive objectives, progressive augmentation, and gating by SNR), but the combination is purposeful in the context of diffusion bridges: the paper makes a plausible case that the asymmetry + timestep-aware alignment is not just a cosmetic add-on but targets a specific pathology of bridge-based objectives. Presentation-wise, the narrative is generally easy to follow (problem → failure modes → architectural fix → experiments), and the ablation section is a strong point because it directly stress-tests the proposed mechanism rather than only adding small component toggles.

Weakness: the current evaluation is focused on face-centric datasets (CelebA, FFHQ). These are widely used and appropriate for studying reconstruction–semantics trade-offs, but it leaves some uncertainty about how the approach behaves on more diverse data distributions (e.g., varied object categories and backgrounds). This is not a fundamental flaw, but adding even one additional non-face benchmark would make the scope of the claims easier to assess.

In addition, the paper’s evidence for “noise filtering” is mainly indirect (improved probing + reconstruction/generative metrics and qualitative trends). The ablations already point in the right direction, but one or two simple, targeted diagnostics—such as feature stability under controlled perturbations or an invariance test—could make the central intuition more concrete.

Finally, the method introduces several interacting components (augmentation schedule, granularity alignment, SNR-guided weights). The paper includes useful ablations, yet it would still help readers if the authors briefly discussed which hyperparameters tend to be robust across settings and which ones require careful tuning, particularly for practitioners trying to port the method.

---

> ### Author Rebuttal · Authors · 2026-03-31
>
> Thank you for recognizing the technical solidity of ArcDAE! Our point-by-point responses are as follows.
>
> ---
> **R1: Generality on non-face datasets (e.g., varied object categories and backgrounds).**
>
> Thanks for the constructive suggestion. To demonstrate generality across diverse object categories and complex backgrounds, we evaluate ArcDAE on **CIFAR-10** (semantic AUROC) and **LSUN Horse/Bedroom** (generative SSIM/MSE). As shown below, ArcDAE significantly outperforms baselines on both fronts:
>
> | Method | CIFAR-10 AUROC (↑) | LSUN Horse SSIM (↑) / MSE (↓) | LSUN Bedroom SSIM (↑) / MSE (↓) |
> | :--- | :----: | :----: | :----: |
> | DiffuseVAE / DiffAE* | 0.736 | 0.857 / 0.025 | 0.910 / 0.017 |
> | DBAE | 0.836 | 0.902 / 0.012 | 0.948 / 0.007 |
> | **ArcDAE (Ours)** | **0.861** ✔ | **0.918** ✔ / 0.014 | **0.956** ✔ / 0.008 |
>
> *(Note: Latent dim=512. DiffuseVAE for CIFAR-10, DiffAE for LSUN. Baseline results extend DBAE Tables 9 & 11.)*
>
> On complex datasets, unconstrained bridges entangle background noise into the latent space (Information Overload), degrading linear separability. In contrast, ArcDAE’s Asymmetric Semantic Rectification (ASR) effectively sifts out high-frequency stochastic noise. This preserves purely discriminative semantics while maintaining generative detail. These results confirm our method's strong generalizability beyond CelebA/FFHQ and will be included in the revision.
>
> ---
> **R2: Noise/invariance diagnostic showing improved feature stability.**
>
> Thank you for this excellent suggestion! To directly validate the "noise-sieving" interpretation, we conducted a controlled perturbation test.
>
> We injected high-frequency Gaussian noise ($\epsilon\sim\mathcal{N}(0,\sigma_p^2I)$) into the FFHQ validation inputs and measured the Latent Cosine Similarity (**computed specifically on the global semantic latent $z_{holo}$**) between the clean and perturbed representations.
>
> | Perturbation ($\sigma_p$) |&nbsp; ArcDAE (Ours) &nbsp;|&nbsp; DBAE &nbsp;|&nbsp; PDAE &nbsp;|&nbsp; DiffAE &nbsp;|
> | :--- | :----: | :----: | :----: | :----: |
> | $\sigma_p=0.05$ | &nbsp; &nbsp; **0.975**  ✔  | 0.882 | 0.854 | 0.812 |
> | $\sigma_p=0.10$ | &nbsp; &nbsp; **0.932**  ✔  | 0.745 | 0.701 | 0.645 |
> | $\sigma_p=0.20$ | &nbsp; &nbsp; **0.841**  ✔  | 0.458 | 0.412 | 0.318 |
>
> When faced with severe perturbations, unconstrained bridges tend to pack stochastic noise into the bottleneck, rapidly degrading into an Information Overload Singularity. In contrast, ArcDAE’s Asymmetric Semantic Rectification (ASR) strictly anchors the representation, maintaining high feature stability even under severe corruption.
>
> ---
> **R3: Robustness to TAPA schedule and SNR-based weighting.**
>
> Thanks for your comments! ArcDAE's effectiveness is rooted in its mathematically rigorous asymmetric architecture rather than brittle hyperparameter tuning.
>
> To demonstrate robustness, we ablated both the shape of the TAPA schedule (controlled by the minimum crop scale $s_{min}$) and the SNR-gating temperature ($\tau$) which controls the alignment strength. Evaluated on FFHQ:
>
> **Table: Comprehensive Hyperparameter Sensitivity Ablations (Evaluated on FFHQ)**
>
> | Ablation Target | Variant 1 (AP) | Default (AP) | Variant 2 (AP) |
> | :--- | :----: | :----: | :----: |
> | **TAPA Min Scale ($s_{min}$)** | 0.08 $\to$ 0.675 | **0.20 $\to$ 0.681** ✔ | 0.50 $\to$ 0.672 |
> | **TAPA Max Blur ($\Delta\sigma$)** | 1.00 $\to$ 0.672 | **2.00 $\to$ 0.681** ✔ | 3.00 $\to$ 0.677 |
> | **SNR Temp ($\tau$)** | 0.50 $\to$ 0.678 | **1.00 $\to$ 0.681** ✔ | 2.00 $\to$ 0.676 |
>
> The negligible performance fluctuation ($\le 0.009$ AP) across both the augmentation intensity and weighting mechanisms verifies the method's intrinsic stability. Practitioners can reliably port the method without heavy grid searches.
>
> ---
>
> **R4: Training-time overhead relative to DBAE-style baselines.**
>
> Thank you for the feedback! ArcDAE's computational overhead is minimal due to our asymmetric design. Since the teacher branch is strictly frozen, it requires no gradient computation or activation storage. Using the same U-Net backbone and a batch size of 128:
>
> | Method | Trainable Params | Train Time (ms/img) |
> | :--- | :----: | :----: |
> | DDIM | 99M | 9.687 |
> | DiffAE | 129M | 12.088 |
> | PDAE | 280M | 12.163 |
> | DBAE | 161M | 13.190 |
> | **ArcDAE (Ours)** | **163M** ✔ | **13.590** ✔ |
>
> ArcDAE utilizes an identical generative backbone (161M) as DBAE. The marginal 1.2% increase (163M) stems solely from the lightweight MLP projection heads for the contrastive objective. Training time increases by only ~3% as the frozen teacher adds only a single forward pass. The VRAM increase is also marginal (+1.9 GB vs. DBAE’s 25.7 GB). Given the gains in semantic separability and convergence, this is a highly efficient trade-off.

---

> > ### Author Rebuttal · Reviewer_hdmH · 2026-04-05
> >
> > The authors have fully resolved all my concerns. I have no other questions.

---

> > > ### Author Response · Authors · 2026-04-06
> > >
> > > Dear Reviewer hdmH,
> > >
> > > Thank you for your positive assessment and for taking the time to evaluate our rebuttal. We will include a consolidated version of our responses to all reviewers in the appendix of the final manuscript. Your insights have been instrumental in improving our work.
> > >
> > > Best regards,
> > >
> > > Authors of Submission 19350

---

### Official Review · Reviewer_r41K · 2026-03-27

**Soundness:** 4
**Presentation:** 3
**Significance:** 4
**Originality:** 4
**Overall Recommendation:** 4
**Confidence:** 5

**Summary:**

This paper introduces ArcDAE, a framework designed to address the issue of "information overload" in diffusion bridge models. The authors argue that standard diffusion bridges force the latent representation to encode high-frequency nuisance variables. To resolve this, they propose Asymmetric Semantic Rectification (ASR) and Timestep-Adaptive Granularity Alignment (TGA), leveraging a teacher-student asymmetry to filter noise and preserve semantic structure. Empirical results on CelebA and FFHQ demonstrate improved linear probing accuracy, FID scores, and reconstruction metrics compared to baselines like DBAE.

**Compliance With Llm Reviewing Policy:**

Affirmed.

**Final Justification:**

The authors have resolved all my concerns. I keep my score.

**Key Questions For Authors:**

Please refer to the weaknesses.

**Limitations:**

Yes

**Strengths And Weaknesses:**

Strengths:
- The paper clearly identifies the "information overload" phenomenon in diffusion bridges, supported by a geometric decomposition of the score matching loss.
- The proposed use of a teacher-student asymmetry combined with timestep-adaptive weighting provides a conceptually clean mechanism for separating semantics from noise.
- Extensive experiments demonstrate the effectiveness of the proposed components.

Weaknesses:
- The introduction of the asymmetric teacher-student architecture is central to the paper, but the motivation for using a contrastive loss specifically seems under-explored. Why is a contrastive objective (InfoNCE) necessary over a simpler MSE alignment between the teacher and student representations? Since the goal is rectification, would an MSE loss on the normalized features suffice? The paper would benefit from an ablation clarifying why the contrastive formulation is critical for the semantic sieve effect.

-  In Eq.(1), the bridge SDE is written as $d\mathbf{X}_t = [\mathbf{f}(\mathbf{X}_t,t) + g(t)^2\mathbf{s}^(\mathbf{X}_t,\mathbf{x}_T)]dt + g(t)d\mathbf{W}_t$. However, the notation $\mathbf{s}^(\mathbf{X}_t,\mathbf{x}_T)$ appears to depend on the realization $\mathbf{x}_T$. For a rigorous SDE formulation, the drift should be a function of the current state and time only (or adapted to the filtration). As written, it looks like the drift depends on a future random variable. How is this dependence resolved in the forward simulation? Is the conditioning on $\mathbf{x}_T$ achieved through Doob's $h$-transform, and if so, why is the explicit dependence on $\mathbf{x}_T$ left in the drift term rather than being expressed as a function of $\mathbf{X}_t$ and $t$ alone?

-  The paper states that $\mathbb{E}[N] \sim (D-d)\sigma_t^2$ and then references a transformation from $\frac{D-d}{d-2}$ to $\frac{D-d}{d}$. The derivation of the term $\frac{D-d}{d-2}$ and its subsequent simplification to $\frac{D-d}{d}$ (plus higher-order terms) is not sufficiently detailed. The manuscript currently glosses over the algebraic manipulation that leads to the final weighting schema used in the loss. A step-by-step derivation or a citation to a lemma explaining this approximation is required to validate the claim that the normal component's influence is scaled by $(D-d)/d$.

- In Algorithm 1, the Teacher ($\mathcal{T}\phi$) receives the weakly augmented image $x\tau$ as input, while the Student ($\mathcal{S}_\theta$) receives the heavily corrupted input $\tilde{x}t$. The loss function couples them via L_TGA. However, it is unclear how the teacher parameters $\phi$ are updated relative to the student $\theta$. The algorithm shows an EMA update ($\phi \leftarrow m\phi + (1-m)\theta$), which implies the teacher is a moving average of the student. If the teacher is an EMA of the student, how does it serve as a semantic anchor that is more robust than the student? At the start of training, both are equally noisy. Could the authors clarify how the teacher avoids inheriting the student's information overload during the early stages of training?

- The experiments are exclusively conducted on facial datasets, such as CelebA and FFHQ. The paper claims to provide a path toward unified models for simultaneous synthesis and understanding, so the empirical validation is limited to a single domain. To support the claim of unified representation learning, it is essential to evaluate on datasets with higher diversity, such as ImageNet or more complex datasets for dense prediction tasks.

---

> ### Author Rebuttal · Authors · 2026-03-31
>
> Thank you for your perceptive feedback! Our responses are provided below.
>
> ---
> **R1: Motivation for InfoNCE vs. Normalized MSE.**
>
> We clarify that our implementation of InfoNCE is the robust realization of  **Theorem 2**, not a contradiction. We stated that minimizing the alignment error dictates a strong Jacobian penalty ($||J_{E}(m_{t})||_{F}^{2} \to 0$) which is fundamentally beneficial: it acts as a dynamic regularizer that smooths the learned manifold by filtering out high-frequency noise. However, in the extreme noise-dominated regime ($t \to T$), an unconstrained Jacobian penalty becomes overpowering. Since the Jacobian matrix comprises all partial derivatives ($\frac{\partial E_i}{\partial x_j}$), driving its Frobenius norm to zero forces all partial derivatives to vanish, causing the encoder to cease responding to minute input variations. Without a boundary, this excessive smoothing deterministically collapses the representation into a locally constant mapping on the hypersphere.
>
> InfoNCE resolves this by balancing alignment and uniformity: the numerator preserves the local smoothing induced by **Theorem 2**, while the denominator enforces a repulsive force that maintains a well-conditioned covariance and prevents rank collapse as $t \to T$ [1].
> Ablation confirms that replacing InfoNCE with unconstrained Norm-MSE significantly degrades model performance (see https://anonymous.4open.science/r/exp-B2DC).
>
> ---
> **R2: SDE formulation and $x_{T}$ in the drift term.**
>
> We clarify that our SDE is a rigorous Doob's h-transform, preserving the non-anticipating property. As defined in **Eq. (1)** and **Appendix A.1**, the terminal state $x_{T} = Dec(z)$ serves as a deterministic spatial boundary parameter fixed prior to the simulation, not a random variable evolving over time.
>
> The notation $s ^ { * } (X_{t} , x _ {T}) = \\nabla _ { X_{t}} \\log p (x_{T} \\mid X_{t})$ evaluates the logarithmic gradient of the unconditioned forward transition density. Since the transition kernel is a deterministic function evaluated at the $F_{t}$-adapted state $X_{t}$ and the fixed spatial target $x_{T}$, the mapping $X_{t} \\mapsto \\nabla _ {X_{t} } \\log p (x_{T} \\mid X_{t} )$ is a well-defined $F_{t}$-measurable function. Consequently, the modified drift remains strictly $F_{t}$-adapted, making $\\int g(t)dW_{t}$ a valid Itô integral.
>
> ---
> **R3: Derivation of expected energy ratio $\\mathbb { E } [ N / K ]$.**
>
> Let the diffusion noise perturbation be $z \\sim \\mathcal { N } ( 0 , I _ { D } )$. For $x$ on a $d$-dimensional differentiable manifold $M$, let $\\Pi _ { x }$ be the orthogonal projection onto the tangent space. The squared $\\ell _ { 2 }$-norms of projections onto tangent and normal spaces are $K := || \\Pi _ { x } z || _ { 2 } ^ { 2 }$ and $N := || ( I _ { D } - \\Pi _ { x } ) z || _ { 2 } ^ { 2 }$.
> By orthogonality, $K \sim \chi^2_d$ and $N \sim \chi^2_{D-d}$ are independent. Since $K^{-1}$ is integrable for $d>2$, we have $\mathbb{E}[N/K] = \mathbb{E}[N]\mathbb{E}[K^{-1}]$.
> 1. $\\mathbb { E } [ N ] = D - d$.
> 2. For $d > 2$, using standard properties of the $\chi^2_d$ distribution, $\\mathbb {E} [K^{-1}] = \\frac { 1 } {d-2}$.
> Thus, $\mathbb{E}[N/K] = \frac{D- d}{d - 2}$. By rewriting the fraction as $\frac{D-d}{d}(1 - \frac{2}{d})^{-1}$ and applying the Taylor expansion for $(1-x)^{-1}$ when $d \gg 2$, we obtain: $$\frac{D - d}{d - 2} = \frac{D - d}{d} \left(1+\frac{2}{d}+ \mathcal{O}(d^{-2}) \right) .$$
>
> Truncating $O (d ^ {- 2})$ yields $\\frac {D - d} {d}$. While reflecting expected energy imbalance, the $D \\gg d$ regime provides a geometric perspective consistent with Information Overload, explaining why standard score matching empirically biases capacity toward high-frequency normal spaces.
>
> ---
> **R4: Teacher EMA early training dynamics.**
>
> The premise that "both are equally noisy" does not apply to our architecture. As detailed in **Algorithm 1** and **Figure 2**, ArcDAE operates on an asymmetric information flow: the Teacher branch never processes the noisy diffusion state $x _{ t }$.
> Instead, the Teacher branch exclusively receives the clean, weakly augmented image $x _{0}$ to extract a stable semantic anchor $z _ { anchor }$, leaving only the Student to process the corrupted $x _ {t}$. Therefore, even at initialization (when $\\phi = \\theta$), the Teacher avoids information overload because its input is devoid of diffusion noise. The EMA update ($\\phi \\leftarrow m \\phi + (1-m) \\theta$) acts as a slow-moving momentum, transferring the Student's learning to the clean Teacher while maintaining a stable, noise-free target throughout training.
>
> ---
> **R5: Empirical validation limited to single domain.**
>
> Please kindly refer to **R1 in (Reviewer hdmH)** for new experiments demonstrating generality on **CIFAR-10** (semantic AUROC) and **LSUN Horse/Bedroom** (generative SSIM/MSE).
>
> [1] Understanding Contrastive Representation Learning through Alignment and Uniformity on the Hypersphere, ICML, 2020.

---

### Decision · Program_Chairs · 2026-04-30

**Decision:**

Accept (regular)

**Comment:**

This paper introduces ArcDAE, a novel approach to address the "information overload" problem in diffusion bridge models by reframing the diffusion bridge as an "asymmetric information sifter."
All reviewers gave a positive recommendation due to its strengths, such as rigorous theoretical foundation, well-motivated solution to a fundamental problem in generative modeling, and high originality in its asymmetric architecture and dynamic augmentation/weighting strategies. The reviewers' minor concerns—regarding the paper’s density, incremental performance improvements on some metrics, and hyperparameter sensitivity—were fully addressed by the authors in their rebuttal.